# Upcycling of atmospheric CO$_2$ to self-healing recyclable polymers under ambient conditions

Xiaoyue Zeng[1,3], Shiguang Zhang[1,3], Huiya Li[2,3], Chun Liu [1] & Liang Chen [1] ✉

The polymer industry is confronting an urgent sustainability trilemma: accelerating plastic pollution, substantial CO$_2$ emissions from production processes, and dependence on diminishing fossil resources. Upcycling CO$_2$ into polymers presents a promising solution to these interconnected issues; however, existing CO$_2$-to-polymer technologies face significant challenges: dependence on concentrated CO$_2$ sources rather than direct air capture (DAC), reliance on complex catalysts and energy-intensive conditions (elevated temperatures/pressures), and generation of polymers with limited self-healing and recyclability. Herein, we propose a catalyst-free strategy of converting atmospheric CO$_2$ into carbonate ions (CO$_3^{2-}$) as intermediates for the synthesis of dynamic covalent polymers. This approach is based on a dynamic bond system, termed the CO$_3^{2-}$-bridged dynamic covalent bond, enabling catalyst-free synthesis of polymers from ambient air at room temperature and pressure. The resultant polymers show excellent mechanical properties, rapid self-healing, and versatile circularity through three distinct pathways: thermal reprocessing, closed-loop chemical recycling via acid-triggered depolymerization at room temperature, and upcycling of mixed CO$_2$-derived polymers into hybrid materials with enhanced properties. This study provides a platform for both low-energy-consuming CO$_2$ valorization and the development of sustainable polymers.

The global plastics industry confronts an escalating environmental pollution crisis[1–3]. Phasing out plastics entirely is neither feasible nor practical, as no alternatives can rival their unparalleled combination of lightweightness, durability, versatility, easy processing and affordability. This dilemma has spurred the development of innovative closed-/open-loop polymer recycling or upcycling to advance the circular economy[4–15]. Notably, dynamic covalent polymer, also known as covalent adaptable network (CAN), has emerged as a promising solution[9–15]. CAN undergoes bond exchange reactions in the solid state, enabling materials to undergo thermal reprocessing similarly to thermoplastics, while retaining the advantageous properties of

thermosets[16,17]. This solid-state exchange reaction also endows materials with inherent self-healing capabilities[18]. Besides, some CANs can be depolymerized back to their precursors through the reversible dissociation of dynamic covalent bonds, thereby facilitating their closed-loop chemical recycling[13,14]. Pioneered by Leibler et al.[9], various dynamic covalent reactions have been harnessed in CAN synthesis, such as imine exchange[19], boronic ester exchange[10], nucleophilic aromatic substitution[14], olefin metathesis[20], diketoenamine exchange[21], vinylogous urethane transamination[22], urethane exchange[23], diselenide exchange[24], oxime-ester exchange[25], silyl ether exchange[26] and so on. Despite these advances, current CAN systems

[1]Chongqing Key Laboratory of Soft-Matter Material Manufacturing, School of Chemistry and Chemical Engineering, Southwest University, Chongqing, China. [2]Analytical & Testing Center, Southwest University, Chongqing, China. [3]These authors contributed equally: Xiaoyue Zeng, Shiguang Zhang, Huiya Li. ✉e-mail: liangchen16@swu.edu.cn

remain challenged by harsh reprocessing conditions (elevated temperatures/pressures, catalyst dependence), limited chemical recyclability, and the persistent strength-toughness trade-off[12,13,23,27,28]. These limitations highlight the urgent need for dynamic chemistries capable of simultaneously achieving superior reprocessability, chemical recyclability, and balanced mechanical properties.

While recycling innovations are vital, sustainable polymer production from renewable feedstocks is also critical. Among various renewable resources, $CO_2$ stands out as a particularly promising feedstock, offering dual benefits of mitigating carbon emissions and reducing dependence on petrochemical resources[29–32]. Unfortunately, the conversion of $CO_2$ into polymers is not without challenges. Current methodologies for this C1 building block conversion rely upon concentrated pure $CO_2$ (Supplementary Tables S1, S2). Direct air capture and utilization (DACU) represents a promising alternative pathway but remains challenging[33], as it demands reaction systems with extraordinary selectivity and efficiency under ultra-dilute atmospheric $CO_2$ conditions (~400 ppm). Furthermore, most current synthetic pathways for $CO_2$-derived polymers require sophisticated catalytic systems, high-energy precursors, and harsh reaction conditions (elevated pressures and temperatures) (Supplementary Tables S1, S2)[27,34–36], thereby compromising economic viability and scalability. Although a few catalyst-free methods have recently emerged, significant hurdles persist in overcoming unfavorable reaction conditions and attaining excellent thermal and mechanical properties[37–40]. Additionally, polymers synthesized via these methods typically exhibit limited recyclability and lack inherent self-healing capacities[41,42].

The fundamental challenge in converting $CO_2$ arises from its inherent stability and inertness, yet it also possesses a slight acidity. Previous studies have demonstrated that alkali hydroxides can efficiently (>90% yield) capture atmospheric $CO_2$ and convert it into carbonate ions ($CO_3^{2-}$)[43]. This inspires us to explore the feasibility of developing a catalyst-free approach to convert atmospheric $CO_2$ into $CO_3^{2-}$ as intermediates for synthesizing self-healing recyclable polymers. To achieve this goal, the key is to identify an appropriate functional group capable of reversibly binding to $CO_3^{2-}$. In this regard, $\alpha,\alpha,\alpha$-trifluoroacetophenone (TFP)—a known ionophore that can form 2:1 tetrahedral adducts with $CO_3^{2-}$ (Fig. 1a)—emerges as a promising candidate[44]. Meanwhile, given its structural similarity to hemiacetal,

we hypothesize that such $CO_3^{2-}$-bridging interactions may exhibit high plasticity, making it an attractive option for constructing recyclable polymers. Here, we show that, unlike conventional synthetic routes, this interaction—termed the $CO_3^{2-}$-bridged dynamic covalent bond—forms at ambient temperature through rapid, quantitative reversible reactions between $CO_3^{2-}$ and TFP without the need for catalysts. Furthermore, this bond not only undergoes rapid, catalyst-free dynamic exchange reactions but can also be reversibly cleaved under mild acidic conditions at ambient temperature. Leveraging this chemistry platform, we synthesized a series of CANs directly using ambient air as a reactant under ambient temperature and pressure conditions, without catalysts. The resulting polymers exhibit tunable mechanical properties ranging from PDMS-like elastomers (with a Young's modulus of 2.8 MPa and an elongation at break of 906%) to polystyrene-like rigid plastics (with a Young's modulus of 1.6 GPa and an ultimate tensile stress of 38.6 MPa), ultrafast self-healing capabilities (98.8% efficiency in 10 minutes at 80 °C), and versatile circularity through three distinct pathways: thermal reprocessing through compression molding or extrusion under mild conditions (more than five cycles without performance degradation), closed-loop chemical recycling via acid-triggered dissociation at ambient temperature (more than 95% recovery), and upcycling of mixed $CO_2$-derived polymer networks into hybrid materials with enhanced properties (Fig. 1b). This study demonstrates an energy-efficient approach for simultaneous atmospheric $CO_2$ valorization and sustainable polymer production.

## Results
### Model reactions
The nucleophilic addition reaction between $CO_3^{2-}$ and TFP has previously garnered attention, with successful integrations into chemical sensors and ion channels[44–48]. However, a comprehensive understanding of this unique reaction, its dynamic behavior, and its unexplored potential in polymer synthesis still remains elusive. To gain insights into the TFP-$CO_3^{2-}$ reaction, we designed tetrabutylammonium carbonate (TBAC) and TFP as the model compounds (Fig. 2a, Supplementary Fig. 1). Upon mixing them in a 2:1 molar ratio in deuterated acetonitrile ($CD_3CN$) and allowing the reaction to proceed for 15 minutes under ambient temperature, both $^1H$ and $^{19}F$ NMR spectroscopies revealed the occurrence of a quantitative reaction with no trace of side

**Fig. 1 | Methodologies for sustainable polymers. a** Reversible TFP-$CO_3^{2-}$ reaction. **b** Conversion of atmospheric $CO_2$ into $CO_3^{2-}$ as intermediates for the catalyst-free synthesis of self-healing recyclable polymers under ambient conditions.

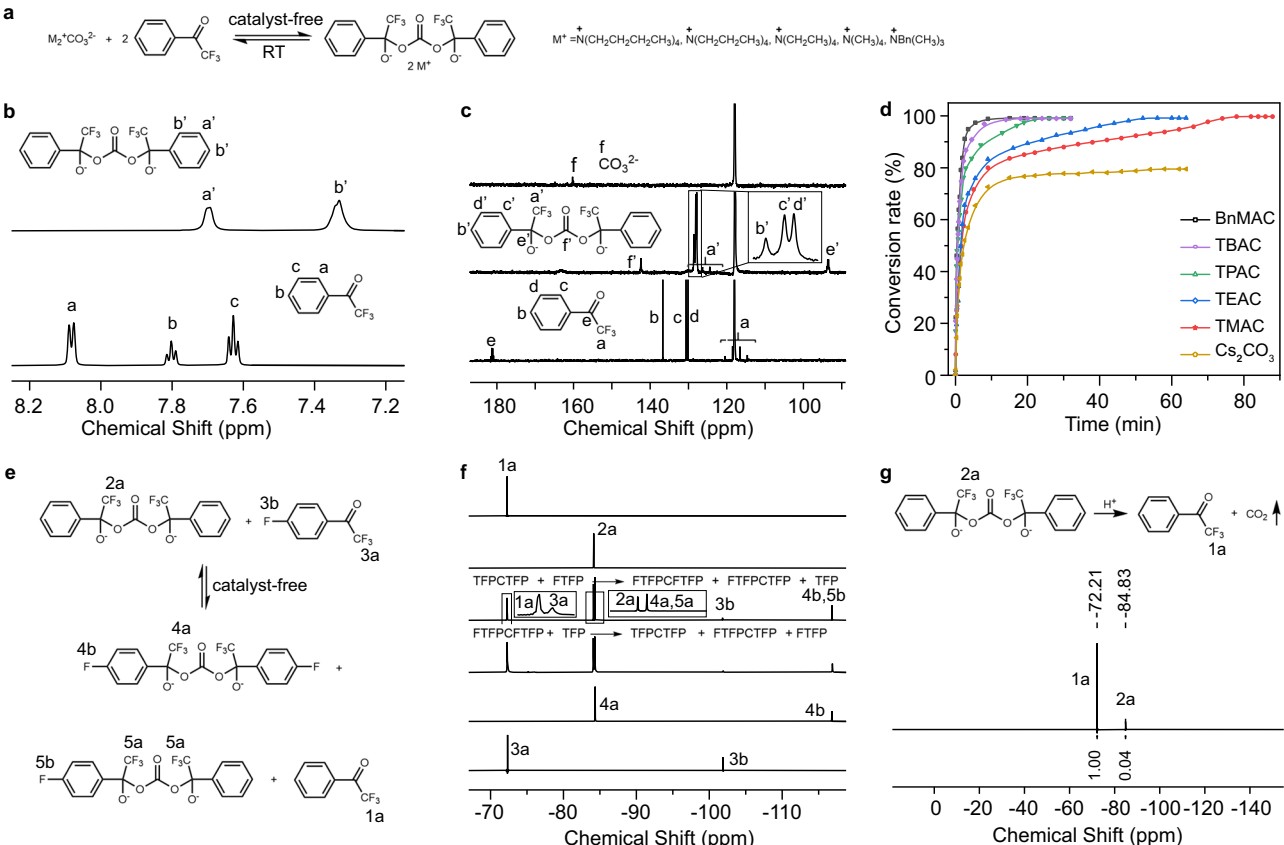

**Fig. 2 | Model reactions for $CO_3^{2-}$-bridged dynamic covalent bond formation, dynamic exchange and dissociation. a** Catalyst-free reaction between $CO_3^{2-}$ and TFP. Quantitative reaction between TBAC and TFP at ambient temperature verification: ¹H NMR spectra (**b**) and ¹³C NMR spectra (**c**). **d** Effect of counter-cations on TFP-$CO_3^{2-}$ reaction kinetics. **e**, **f** Catalyst-free exchange reaction and ¹⁹F NMR spectra comparison (from top to bottom): TFP, TFPCTFP, equilibrium state of exchange reaction started from TFPCTFP and FTFP, equilibrium state of exchange reaction started from FTFPCFTFP and TFP, FTFPCFTFP, and FTFP. **g** Acid-induced dissociation reaction and corresponding ¹⁹F NMR spectra. All NMR spectra were acquired in CD₃CN.

reactions in the absence of catalysis, as evidenced by the completely upfield shifts in the proton signals of the benzene ring (Fig. 2b and Supplementary Figs. 2, 3, $H_a$ 8.08 → $H_{a'}$ 7.69 ppm, $H_b + H_c$ 7.80 + 7.63 → $H_{b'}$ 7.33 ppm) and fluorine signals (Supplementary Fig. 4, $F_a$ −72.27 → $F_{a'}$ −84.15 ppm). A more detailed analysis of this reaction was performed using ¹³C NMR spectra. As depicted in Fig. 2c, distinct shifts were observed in the carbon signals of the trifluoromethyl group (quartet peak, $C_a$ 118.01 → $C_{a'}$ 126.29 ppm) and the benzene ring ($C_{b-d}$, 136.68, 130.64 and 130.14 → $C_{b'-d'}$ 128.59, 128.08, and 127.86 ppm). Meanwhile, the carbonyl carbon peak of TFP at 181.11 ppm and the carbonate-related carbon peak at 160.26 ppm completely disappeared, replaced by new peaks at 93.44 ppm and 142.31 ppm, respectively, in the product spectrum. These findings suggest a nucleophilic attack by $CO_3^{2-}$ on the carbonyl carbon of TFP occurs, leading to the formation of anionic $CO_3^{2-}$-bridged adducts. This result was further confirmed by Fourier-transform infrared (FTIR) spectra and mass spectroscopy (MS) analysis (Supplementary Figs. 5-6).

To investigate the influence of counter-cations on the reaction, both organic and inorganic carbonate salts were employed. In the case of organic carbonates, including TBAC, tetrapropylammonium carbonate (TPAC), tetraethylammonium carbonate (TEAC), tetramethylammonium carbonate (TMAC), and benzyltrimethylammonium carbonate (BnMAC), all exhibited near-quantitative yields while demonstrating marked variations in reaction rates (Fig. 2d, Supplementary Figs. 7–10, 12a–e). The reaction rates decreased in the following order: BnMAC > TBAC > TPAC > TEAC > TMAC. Notably, even TMAC, which exhibited the slowest rate, reached equilibrium within 80 minutes. In contrast, inorganic carbonates displayed significantly

lower conversion rates (Supplementary Fig. 11, Fig. 12f), presumably due to their poor solubility and elevated dissociation energies in organic solvents. These findings indicate the advantage of organic carbonate salts for the rapid and efficient preparation of $CO_3^{2-}$-bridged adducts.

The formed $CO_3^{2-}$-bridged bond exhibited remarkable dynamic exchange properties. When stoichiometric mixing of competitive reactant 4-fluoro-α,α,α-trifluoroacetophenone (FTFP) with $CO_3^{2-}$-TFP adducts (TFPCTFP) in CD₃CN was followed by 10-minute incubation at ambient temperature, five expected compounds were detected via ¹H, ¹⁹F NMR spectroscopy and MS (Fig. 2e-f and Supplementary Figs. 13, 14). This finding confirmed the occurrence of rapid exchange reactions in the absence of catalysts. The dynamic process was further validated by an analogous equilibrium in the reaction between $CO_3^{2-}$-FTFP adducts (FTFPCFTFP) and TFP. Beyond dynamic exchange capabilities, the $CO_3^{2-}$-bridged bond exhibited efficient acid-triggered cleavage under mild conditions. Upon adding aqueous H₂SO₄ to TFPCTFP solution in CD₃CN to achieve pH 4, abundant emission of CO₂ gas was detected (Supplementary Fig. 15 and Movie 1). Subsequent ¹H and ¹⁹F NMR analysis of the residual solution revealed that TFPCTFP decomposed into TFP with a yield of 96.2% (Fig. 2g and Supplementary Fig. 16). Collectively, these results establish that $CO_3^{2-}$-bridged bonds are formed readily, undergo rapid dynamic exchanges, and can be decomposed efficiently under mild conditions.

## Synthesis and characterization of CANs
Motivated by the results of the small-molecule model reaction, we next explored the potential for developing recyclable polymers. Since CO₂ can be captured from air by alkali hydroxide and converted into

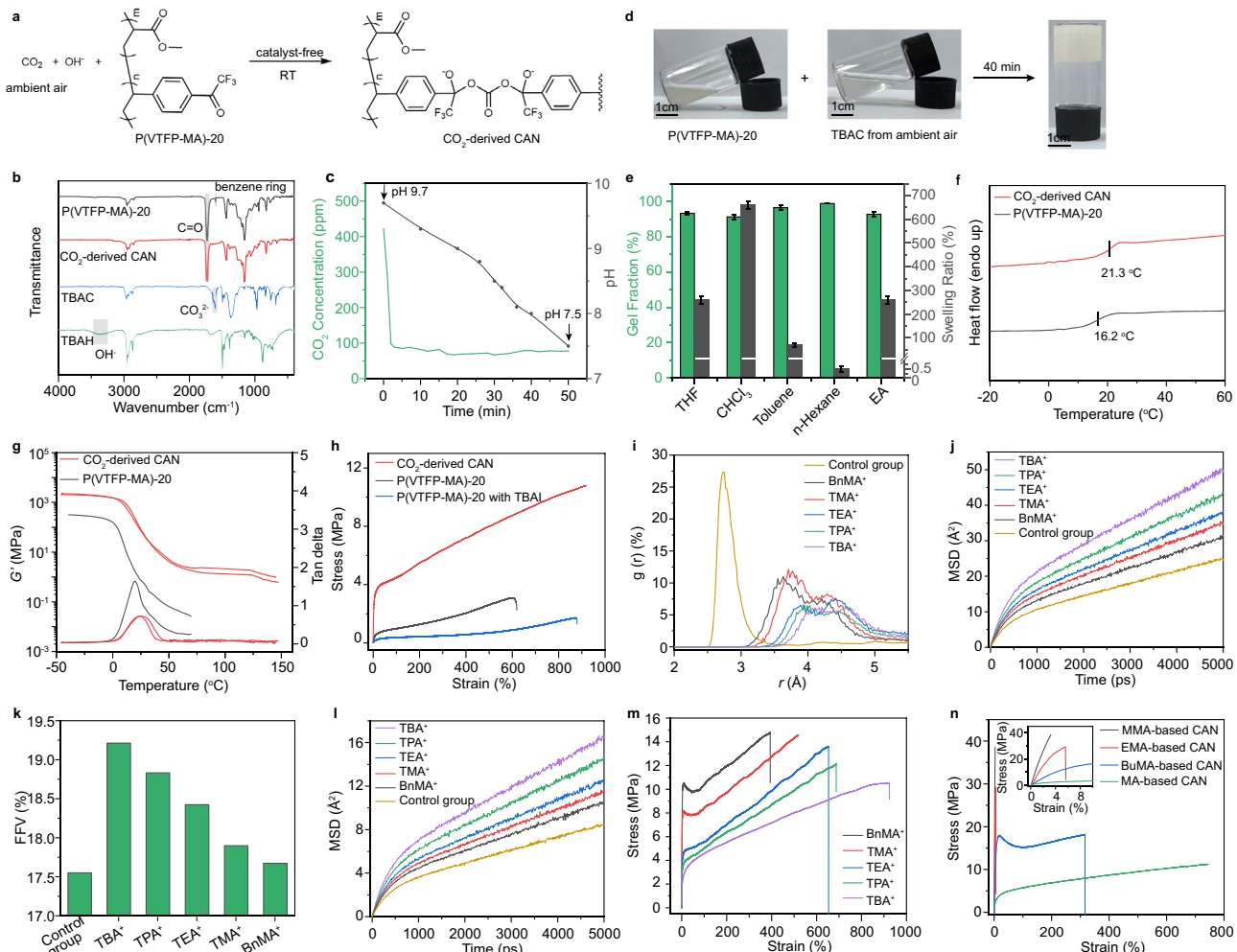

**Fig. 3 | Synthesis and characterization of CANs. a** Schematic illustration of the catalyst-free conversion of atmospheric CO₂ into CANs. **b** FTIR spectra of TBAH, TBAC, P(VTFP-MA)−20 and the resulting CO₂-derived CAN. **c** CO₂ concentration in the exhaust stream and pH of the solution over time during the bubbling of ambient air. **d** Photographs showing the gelation process. **e** Gel fraction and swelling ratio of the CO₂-derived CAN in various solvents. DSC curves (**f**) and DMA curves (**g**) of P(VTFP-MA)−20 and the CO₂-derived CAN. **h** Tensile stress-strain curves of P(VTFP-MA)−20, CO₂-derived CAN and P(VTFP-MA)−20 with TBAI additive. **i** $g(r)$ as a function of O⁻·N⁺ distance $r$. **j** MSD of counter-cations over time. **k** FFV of CANs. **l** MSD of polymer backbone over time. Tensile stress-strain curves of CANs with varying counter-cations (**m**) and linear copolymers (**n**). Data are presented as mean ± standard deviation of three independent experiments.

$CO_3^{2-}$ [43], our reversible chemical platform enables catalyst-free synthesis of recyclable polymers from atmospheric air via $CO_3^{2-}$ as intermediates. As a proof-of-concept, we synthesized the linear copolymers poly(4-vinyltrifluoroacetophenone-co-methyl acrylate) [P(VTFP-MA)−20, VTFP/MA = 1/20 mol/mol, $M_n$ = 48.1 kDa, Đ = 1.53] via free-radical copolymerization of 4-vinyltrifluoroacetophenone (VTFP) and methyl acrylate (MA) monomers (Fig. 3a and Supplementary Figs. 17–24). The VTFP monomer was prepared in 85% isolated yield through a one-step Suzuki coupling of commercially available reagents, underscoring its potential for scalable production. Subsequently, a one-pot, two-step method was designed to synthesize polymer networks from atmospheric CO₂. In the first step, ambient air from Chongqing, China (29° 49′ 20.68″ N, 106° 25′ 23.0″ E) was bubbled at ~300 mL·min⁻¹ into an acetonitrile (CH₃CN)/H₂O solution of tetrabutylammonium hydroxide (TBAH) under ambient conditions, yielding TBAC (Fig. 3b and Supplementary Figs. 25, 26). Monitoring the CO₂ concentration in the exhaust stream revealed that it did not rise significantly until complete conversion of TBAH to TBAC (Fig. 3c), with the endpoint determined by the solution pH reaching ~7.5 −a value corresponding to that of a pure TBAC solution at the same concentration. Quantitative

analysis confirmed the TBAC purity of 94.1% (containing ~5.9% tetrabutylammonium hydrogen carbonate (TBAHC), which was verified to not affect subsequent cross-linking reaction or material properties; Supplementary Figs. 27, 28) and an atmospheric CO₂ capture efficiency (defined as the mass ratio of fixed CO₂ to supplied CO₂) of 80.0%, demonstrating the high efficacy of CO₂ fixation. The cross-linked network was then prepared by adding P(VTFP-MA)−20 solution to the as-prepared TBAC solution. Upon mixing, the transparent solution transformed into a white paste within 30 seconds, and subsequently evolved into a semitransparent, free-standing gel after 40 minutes of stirring (Fig. 3d), confirming successful cross-linking. After solvent removal, the resulting CO₂-derived CAN (unless otherwise specified, CO₂-derived CAN is defined in this work as the network derived from P(VTFP-MA)−20, TBAH, and ambient air) was hot-pressed at 90 °C and 6 MPa for 10 minutes, producing a slightly colored, transparent CAN film exhibiting a transmittance exceeding 70% in the visible light region from 500 to 800 nm (Supplementary Fig. 29). Swelling tests confirmed the film's insolubility in various organic solvents (e.g., tetrahydrofuran (THF), chloroform (CHCl₃), toluene, n-hexane, ethyl acetate (EA)) after 12 h immersion, with gel fractions exceeding 90% in

all cases, further verifying the cross-linked structure (Fig. 3e). FTIR spectroscopy confirmed the formation of $CO_3^{2-}$-bridged networks, as evidenced by the disappearance of detectable carbonate signals and a concurrent reduction in the benzene ring vibration of P(VTFP-MA) −20 (Fig. 3b).

Having successfully synthesized the CAN, we proceeded to investigate their thermal and mechanical properties. Differential scanning calorimetry (DSC) revealed a glass transition temperature ($T_g$) of 21.3 °C for the $CO_2$-derived CAN (Fig. 3f), which was higher than that of its linear polymer precursor. This elevated $T_g$ was further corroborated by dynamic mechanical analysis (DMA, Fig. 3g), which showed a characteristic drop in the storage modulus ($E'$) at the transition. Critically, a distinct rubbery plateau in $E'$ above $T_g$ was observed for the $CO_2$-derived CAN, whereas no such plateau was evident for the P(VTFP-MA)−20, confirming the crosslinked nature of the network[49]. The crosslink density ($v$) was calculated to be 79.1 mol m$^{-3}$ from the rubbery plateau modulus ($E'_r$) using the equation $v = E'_r/[3R(T_g + 50)]$. Thermogravimetric analysis (TGA) demonstrated excellent thermal stability, with a 5% mass loss temperature ($T_{5\%}$) of 252 °C (Supplementary Fig. 30). To evaluate the mechanical properties, uniaxial tensile tests were conducted. The results indicated that the ultimate tensile stress, Young's modulus, toughness, and elongation at break of the crosslinked CANs were 10.7 MPa, 84.6 MPa, 68.7 MJ·m$^{-3}$, and 937%, respectively (Fig. 3h and Supplementary Fig. 31). Compared to the linear polymer P(VTFP-MA)-20, these values demonstrated respective increases of 3.45-, 32.38-, 5.81-, and 1.56-fold. This exceptional ability to simultaneously enhance strength, rigidity, toughness, and ductility is unusual, as these properties are often seen as conflicting[50,51]. We suspect that this paradoxical mechanical behavior may be attributed to the unique dual role of the ionic crosslinker, where $CO_3^{2-}$ ions function as crosslinkers, enhancing strength and stiffness, while the counter-cations serve as plasticizers, increasing toughness and ductility. The plasticizing effect of the counter-cations was initially confirmed in a control experiment, where blending tetrabutylammonium iodide (TBAI) with the linear polymer P(VTFP-MA)-20 enhanced ductility but compromised strength and stiffness compared to the pure linear polymer. To gain deeper mechanistic insights, we conducted molecular dynamics (MD) simulations using LAMMPS (Supplementary Figs. 32, 33). A comparative analysis between the $CO_2$-derived CAN (with TBA$^+$) and the control system (with H$^+$ replacing TBA$^+$) revealed key distinctions: (i) a weaker ionic association, as indicated by the significantly right-shifted first peak in the radial distribution function (RDF, g(r)) between TBA$^+$ and the polymeric anions (Fig. 3i); (ii) significantly enhanced cation mobility, with a diffusion coefficient 2.1 times greater than that of the control group (Fig. 3j and Supplementary Fig. 34a); (iii) a considerably expanded polymer free volume fraction (FFV, Fig. 3k and Supplementary Fig. 35), which provides more space for segmental motion; and (iv) dramatically accelerated relaxation dynamics of the polymer chains, as demonstrated by a significantly increased mean-squared displacement (MSD) over time (Fig. 3l and Supplementary Fig. 34b). These findings collectively demonstrate that the bulky, flexible TBA$^+$ cations effectively screen electrostatic interactions, enhance ion mobility, expand free volume, and lubricate polymer chains, thereby rationalizing the enhanced ductility and toughness.

This cation-mediated plasticization enables precise mechanical tailoring through cation selection. As shown in Fig. 3m, systematically shortening the alkyl chain length from TBA$^+$ to TMA$^+$ led to increases in Young's modulus and ultimate tensile strength but at the expense of ductility. This trend was further accentuated with rigid aromatic counter-cations such as BnMA$^+$, which yielded the highest Young's modulus of 546 MPa—6.5 times that of the TBA$^+$-based CAN—alongside a further reduced elongation at break. MD simulations attribute this systematic mechanical variation to a gradual attenuation of the cationic plasticizing effect. RDF analysis showed a progressive left-shift

of the first peak for the cations-polymer interaction along the series TBA$^+$ → TPA$^+$ → TEA$^+$ → TMA$^+$ → BnMA$^+$ (Fig. 3i), indicating a decreasing average ion-pair distance and thus stronger electrostatic interactions. This trend leads to a concomitant decrease in cation mobility and a reduction in polymer free volume (Fig. 3j, k and Supplementary Fig. 34a). As a result, segmental motion of the polymer chains becomes more restricted (Fig. 3l and Supplementary Fig. 34b), shifting the mechanical behavior from ductile to rigid and strong. To further broaden the range of accessible mechanical properties, we modulated the linear polymer backbone. By adjusting comonomer ratios and types, we synthesized a series of CANs whose mechanical properties spanned from flexible elastomers (Young's modulus: 2.8 MPa; elongation at break: 906%) to rigid plastics (Young's modulus: 1.6 GPa, ultimate tensile stress: 38.6 MPa), as shown in Fig. 3n and Supplementary Figs. 37–39. These results indicate the immense potential of our CANs in offering materials with customizable mechanical properties to cater to a diverse range of application requirements. Beyond tunable mechanics, the CANs exhibited exceptional durability. Taking the $CO_2$-derived CAN (selected as the representative system for subsequent investigations unless otherwise specified) as an example, it could withstand 14-day ultraviolet (UV) exposure, high humidity (45% relative humidity), 9 thermal cycles (−20 to 110 °C), and 8-week outdoor exposure with negligible degradation in its mechanical performance, appearance, or chemical structure (Supplementary Figs. 40–43), underscoring its potential for real-world applications.

## Self-healing and reprocessing properties

Although the CANs are crosslinked polymers with robust mechanical properties, they exhibit thermoplastic-like malleability due to catalyst-free exchange reactions. Stress relaxation measurements of $CO_2$-derived CAN conducted at 60 °C and 1% strain (within the linear viscoelastic regime of 0.001%-10%; Supplementary Fig. 44) revealed rapid bond exchange kinetics, with a relaxation time ($\tau$) of only 13.5 seconds (Supplementary Fig. 45). This ultrafast relaxation hindered reliable quantification at elevated temperatures via stress relaxation[49]; we therefore employed small-amplitude oscillatory shear (SAOS). The relaxation time $\tau$ was determined from the crossover frequency ($\omega_c$) where the storage modulus ($G'$) equals the loss modulus ($G''$), using the equation $\tau = 1/\omega_c$ (Fig. 4a). Arrhenius analysis showed a linear correlation between ln($\tau$) and 1000/$T$ (Fig. 4b), confirming the vitrimeric nature of the networks. The activation energy ($E_a$) for the solid-state bond exchange, calculated from the Arrhenius plot, was 111.9 kJ·mol$^{-1}$. This value is closely consistent with the $E_a$ derived from time-temperature superposition (TTS) analysis (Supplementary Fig. 46). Temperature-sweep rheological tests revealed a sharp, nearly three-order-of-magnitude decrease in $G'$, accompanied by a distinct $G'$/ $G''$ crossover at 85 °C (Fig. 4c), consistent with a dissociative exchange mechanism in the $CO_2$-derived networks. To further elucidate this mechanism, we performed in situ variable-temperature FTIR spectroscopy. Upon heating from 25 to 120 °C, a gradual intensification of the absorption band at 1730 cm$^{-1}$ was observed, suggesting the thermally induced cleavage of $CO_3^{2-}$-bridged bonds to form the TFP compound (Fig. 4d). Subsequent cooling at room temperature for 24 hours restored the original $CO_3^{2-}$-bridged bonds, confirming the reversibility. These findings collectively demonstrate the exceptional malleability and rapid dissociative exchange reaction kinetics of the $CO_2$-derived CAN.

Due to the rapid exchange reaction, the CANs exhibit remarkable self-healing capability after mechanical damage. As shown in Fig. 4e, two rectangular $CO_2$-derived CAN samples (30 mm × 10 mm × 0.5 mm) were prepared, with one stained red using rhodamine B for enhanced visibility. After being cut in half, the separated samples were seamlessly reintegrated within 10 minutes at 80 °C under 0.5 MPa pressure, leaving no visible scratches. The repaired sample was capable of supporting a 1 kg object weighing over 5000 times its own weight for at

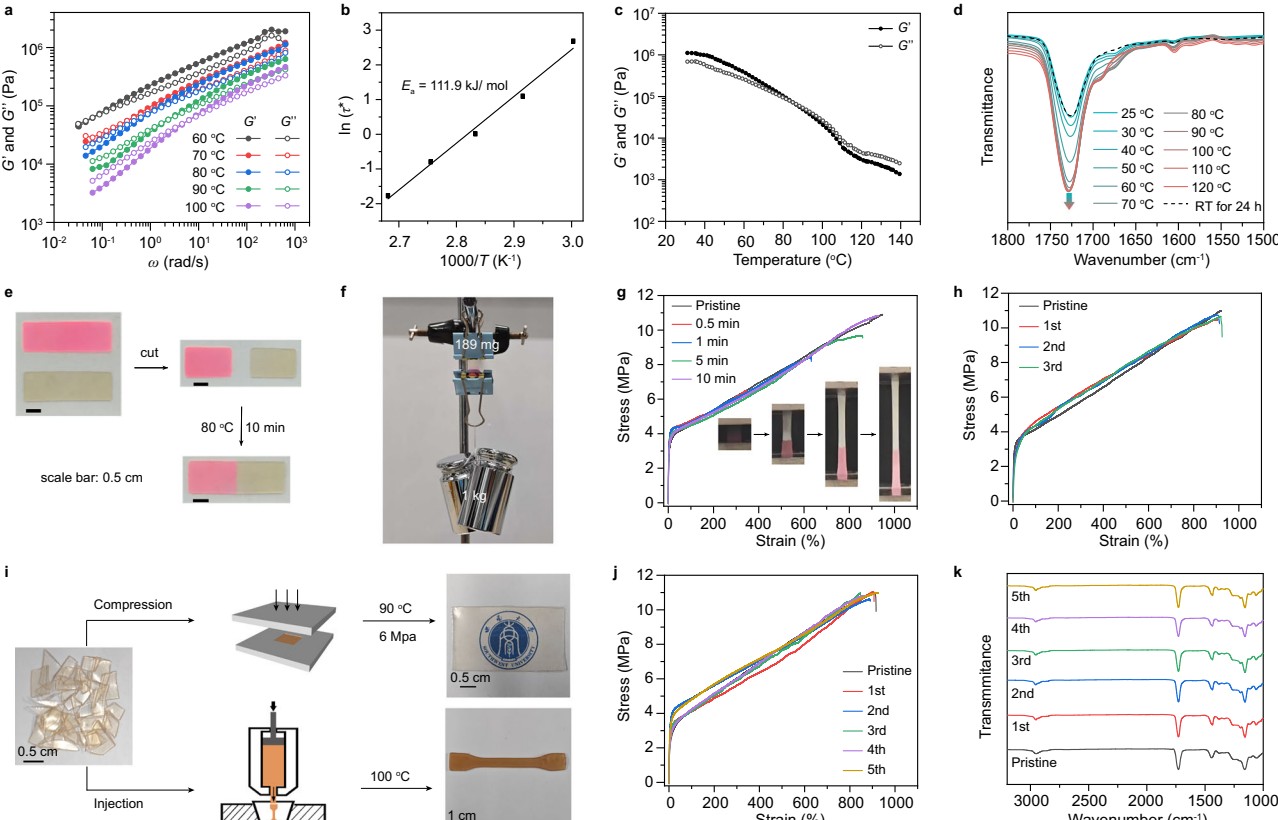

**Fig. 4 | Self-healing and reprocessing properties of CO₂-derived CAN.** SAOS experiments at various temperatures (**a**) and corresponding Arrhenius plot (**b**). **c** Temperature-dependent rheological behavior of the CO₂-derived CAN. **d** In situ variable-temperature FTIR spectra of CO₂-derived CAN. Photographs illustrating the repair of a cut rectangular sheet (**e**) and the repaired sample supporting a weight of up to 1 kg without extension (**f**). **g** Stress-strain curves of samples healed for different durations. *Inset:* Photographic sequence of a 10-minute healed sample during tensile testing. **h** Stress-strain curves of CO₂-derived CAN after three consecutive damage-healing cycles. **i** Photographs of fragmented and reprocessed samples via compression molding and injection molding techniques. Comparative stress-strain curves (**j**) and FTIR spectra (**k**) of the pristine and repeatedly reprocessed CANs via hot-pressing.

least 5 minutes at ambient temperature with negligible extension (Fig. 4f and Supplementary Movie 2). To quantify the self-healing property, we compared the mechanical properties of the original and repaired samples after different healing durations (Fig. 4g, Supplementary Fig. 47 and Supplementary Movie 3). After just 30 seconds of healing, the healing efficiency—defined as the ratio of the ultimate tensile stress of the repaired sample to that of the original—reached 59%. Within 10 minutes, the tensile property of the repaired sample recovered to 98.8% of its initial value. This damage-healing cycle was repeatable at least three times (Fig. 4h and Supplementary Fig. 48). Furthermore, efficient self-healing was also achieved under milder conditions, albeit with longer healing times. After 24 hours at 40 °C under 2 MPa, a healing efficiency of 98.2% was attained; even at room temperature under the same pressure, the efficiency reached 53.2% (Supplementary Fig. 49).

The rapid exchange reaction upon heating also endows CANs with exceptional reprocessability. Fragmented CO₂-derived CAN can be regenerated into monolithic materials through either compression molding at 90 °C and 6 MPa for 10 minutes or injection molding at 100 °C (Fig. 4i and Supplementary Fig. 50). This demonstrates that the CANs can be reprocessed using multiple industrially relevant techniques within a short period under mild conditions. This contrasts sharply with most studied CANs, which typically require prolonged exposure to high pressure and/or temperature, often resulting in unwanted side reactions[52]. To further investigate their reprocessability, we conducted multiple fragment-hot pressing molding treatments. The results show that, even after five cycles, their thermal and

mechanical properties, as well as chemical structure, remain virtually identical to those of the initial samples (Fig. 4j, k and Supplementary Figs. 51, 52).

## Closed-loop chemical recycling and upcycling

After elucidating the robustness and reprocessability of the polymer networks, we embarked on an investigation into their chemical recyclability. As a demonstration, CO₂-derived CAN sheets were fragmented and immersed in a 0.2 M H₂SO₄/ CH₃CN solution at ambient temperature. After a 2.5-hour incubation period, all samples completely dissociated into soluble components (Supplementary Fig. 53a). Increasing the acid concentration to 0.5 M shortened the dissolution time to 40 minutes, yielding a transparent solution without visible residues (Supplementary Fig. 53b). Subsequent liquid-liquid extraction with ethyl acetate and water directly afforded P(VTFP-MA)-20 in ~95% yield without further purification (Fig. 5a and Supplementary Fig. 54). TBAH recovery via basic anion exchange resins achieved 97% efficiency (Supplementary Fig. 55). As expected, the recycled P(VTFP-MA)-20 and TBAH were used to resynthesize CANs. Figure 5b, c and Supplementary Fig. 56 illustrate that the reborn CANs exhibit virtually identical chemical structures, thermal and mechanical properties to those of the original CANs after three cycles of recycling. To demonstrate the possibility of selectively recycling, two scenarios were presented: (i) a mixed plastic waste stream containing CO₂-derived CAN, polyvinyl chloride, polypropylene and polyethylene (Fig. 5d); (ii) a carbon-fiber-reinforced polymer (CFRP) consisting of CO₂-derived CAN and carbon fiber (CF) woven fabrics (Fig. 5e, f and Supplementary Fig. 57).

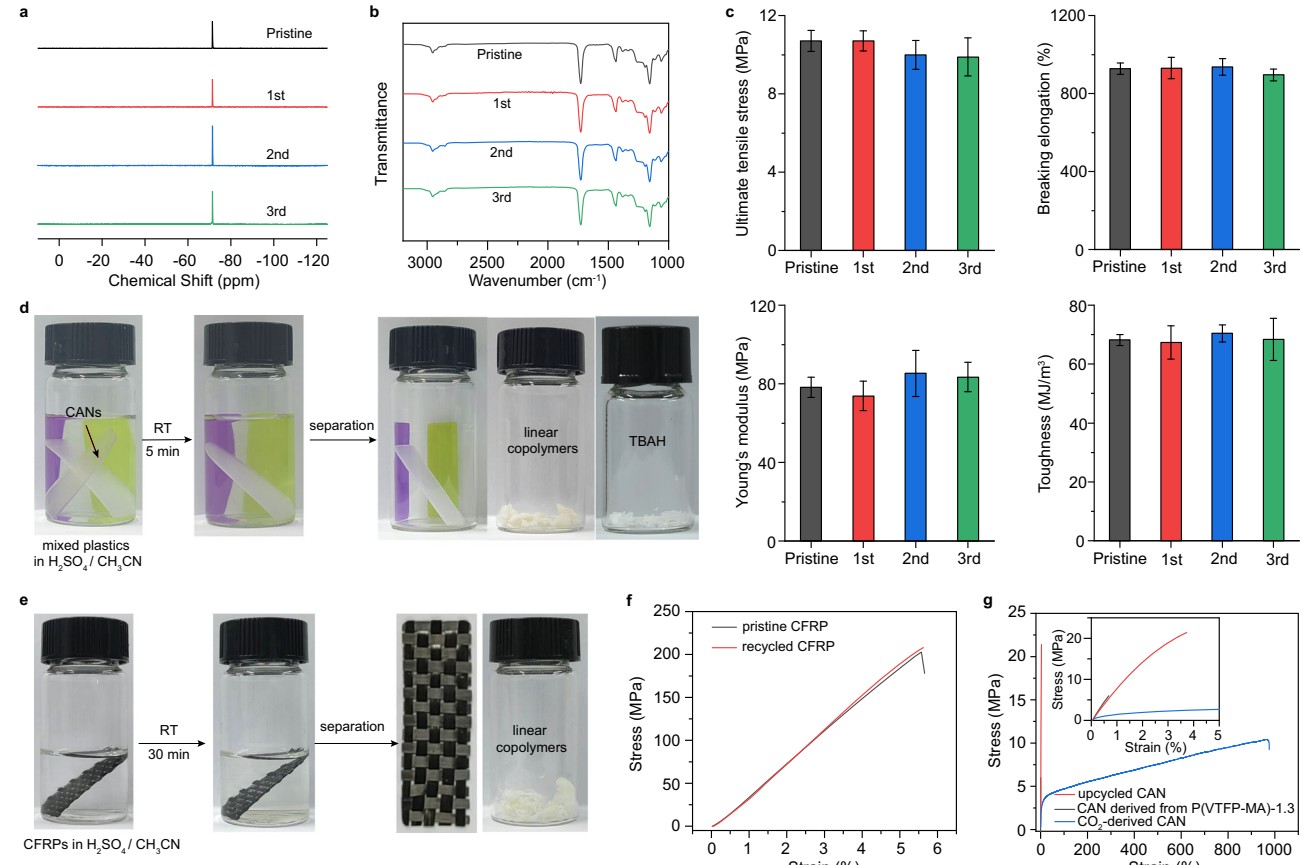

**Fig. 5 | Chemical recycling and upcycling of CO₂-derived CANs. a** Comparison of ¹⁹F NMR spectra (in CD₃CN) of the pristine and recycled P(VTFP-MA)−20. Comparison of chemical structure (**b**) and mechanical properties (**c**) of original and reborn CANs. **d** Photographs showing selective chemical recycling of CANs from a mixed plastic waste stream containing CO₂-derived CAN (transparent), polyvinyl chloride (purple), polypropylene (yellow) and polyethylene (white). **e** Photographs showing selective chemical recycling of CO₂-derived CAN from a CFRP. **f** Stress-strain curves of original and reborn CANs-based CFRP. **g** Stress-strain curves of upcycled CANs, CAN derived from P(VTFP-MA)-1.3, and CO₂-derived CAN. Data are presented as mean ± standard deviation of three independent experiments.

The results show that our mild recycling conditions caused no degradation to non-target materials, indicating the remarkable efficiency and selectivity in degrading, separating, and recovering CANs from complex mixtures.

In addition to closed-loop chemical recycling, the recycled raw materials can also be upcycled into new CO₂-derived polymers. For instance, two distinct CANs were selected—one rigid but brittle (CAN derived from P(VTFP-MA)-1.3), and the other soft yet tough (CO₂-derived CAN). After individual acid-induced degradation into raw materials, they were blended and re-cross-linked via CO₂ introduction, producing a hybrid network with mechanical properties surpassing both original CANs (Fig. 5g and Supplementary Fig. 58). Specifically, the hybrid exhibited 2-fold higher tensile strength and 150-fold enhanced toughness compared to original CANs. These findings demonstrate that the CO₂-derived CAN exhibits both closed-loop chemical recycling and upcycling capabilities under mild conditions.

## Discussion

Existing CO₂-to-polymer methods face the challenge of simultaneously integrating the following desirable traits: rapid and efficient conversion without catalysts under mild conditions, direct utilization of atmospheric CO₂ as a raw material, self-healing and recyclability in the resulting polymers, as well as scalability for large-scale production. Here we present a dynamic covalent bond system—the CO₃²⁻-bridged dynamic covalent bond—and develop a strategy of converting atmospheric CO₂ into CO₃²⁻ as intermediates for the ambient temperature/pressure, catalyst-free synthesis of polymer networks. The resulting polymers exhibit robust mechanical properties, rapid self-healing, reprocessability, and chemical recyclability. This study addresses long-standing challenges in DACU, advancing both C1 chemistry and sustainable polymer science. Meanwhile, we note that the CO₂ content in the current CANs remains limited, ranging from 0.84 to 3.76 wt% (Supplementary Tables S1, S2). Ongoing and future research will focus on optimizing the network design and refining synthetic routes to further enhance CO₂ uptake and improve the thermal and mechanical properties of the resulting polymers. Even at this early stage, this methodology establishes a versatile platform for the development of air-captured CO₂-derived materials.

## Methods

### Materials

TFP, FTFP, benzyltrimethylammonium hydroxide (BnMAH, 25 wt% in water), TBAI (97%), 4-bromo-α,α,α-trifluoroacetophenone (98%), pinacol vinylboronate (97%), TBAHC(95%), cesium carbonate (Cs₂CO₃) were purchased from Bide Pharmatech Ltd. TBAH (40 wt% in water), tetramethylammonium hydroxide (TMAH, 25 wt% in water), tetraethylammonium hydroxide (TEAH, 25 wt% in water), tetrapropylammonium hydroxide (TPAH, 40 wt% in water), 2,2′-Azobis(2-methylpropionitrile) (AIBN, 98%), tetrakis(triphenylphosphine)palladium (Pd(PPh₃)₄, 99%), potassium carbonate (K₂CO₃, 99%), MA (99%), methyl methacrylate (MMA, 99%), ethyl methacrylate (EMA, 99%), butyl methacrylate (BuMA, 99%), 1,4-dioxane, toluene, THF, CH₃CN, n-hexane were acquired from Innochem Co., Ltd. CHCl₃, H₂SO₄, petroleum ether (PE), EA, dichloromethane, methanol (MeOH) were

obtained from Chongqing Taixin Chemical Co., Ltd. All chemical reagents were used as received unless otherwise noted.

## Model reaction for the catalyst-free formation of $CO_3^{2-}$-bridged dynamic covalent bond

The $CO_3^{2-}$-bridged bond formation was investigated using TFP and TBAC as model compounds. TBAC was synthesized in two steps. A TBAH aqueous solution (10 mL, 40 wt%) was saturated with $CO_2$ (99.99%) at room temperature until the pH stabilized, yielding TBAHC. Following the addition of a stoichiometric amount of TBAH (10 mL), the mixture was stirred at ambient temperature for 30 min. The product TBAC was obtained as a white powder by freeze-drying and stored under nitrogen. [1]H NMR (600 MHz, CDCl$_3$) δ 3.31 - 3.17 (m, 8H), 1.60 (qt, $J$ = 8.4, 6.0, 5.1 Hz, 8H), 1.40 (h, $J$ = 7.5, 7.0 Hz, 8H), 1.00 - 0.90 (m, 12H).[13]C NMR (150 MHz, CD$_3$CN) δ 160.59, 59.28, 24.33, 20.30, 13.79. Elemental analysis: calcd for C$_{33}$H$_{72}$N$_2$O$_3$: C/N = 14.14 (w/w), found: C/N = 14.21 (w/w).

For the model reaction, a solution was prepared by mixing TBAC (13.6 mg, 0.025 mmol, 1 equiv.) and TFP (8.7 mg, 0.05 mmol, 2 equiv.) 0.6 mL in CD$_3$CN or CH$_3$CN. After reacting at room temperature for 15 min, the resultant solution was directly characterized by [1]H, [19]F, and [13]C NMR spectroscopy, IR spectroscopy, and mass spectrometry without purification.

## Effect of counter-cations on the formation of $CO_3^{2-}$-bridged dynamic covalent bond

To investigate the influence of counter-cations, a series of carbonate salts was prepared using a method analogous to that for TBAC.

TPAC: [1]H NMR (600 MHz, CD$_3$CN) δ 3.13-3.04 (m, 8H), 1.70-1.59 (m, 8H), 0.93 (q, $J$ = 6.7, 6.0 Hz, 12H). [13]C NMR (150 MHz, CH$_3$CN) δ 160.84, 60.88, 15.98, 10.79. Elemental analysis: calcd for C$_{25}$H$_{56}$N$_2$O$_3$: C/N = 10.71 (w/w), found: C/N = 10.95 (w/w).

TEAC: [1]H NMR (600 MHz, CD$_3$CN) δ 3.22 (q, $J$ = 7.3 Hz, 8H), 1.22 (ddt, $J$ = 7.3, 3.8, 1.9 Hz, 12H). [13]C NMR (150 MHz, CD$_3$CN) δ 161.17, 118.26, 53.03, 53.01, 52.99, 7.67, 1.21. Elemental analysis: calcd for C$_{17}$H$_{40}$N$_2$O$_3$: C/N = 7.29 (w/w), found: C/N = 7.71 (w/w).

TMAC: [1]H NMR (600 MHz, D$_2$O) δ 3.17 (s, 12H). [13]C NMR (150 MHz, D$_2$O) δ 168.27, 55.30, 55.27, 55.24. Elemental analysis: calcd for C$_9$H$_{24}$N$_2$O$_3$: C/N = 3.86 (w/w), found: C/N = 3.85 (w/w).

BnMAC: [1]H NMR (600 MHz, D$_2$O δ 7.60 (ddt, $J$ = 8.5, 5.7, 3.0 Hz, 1H), 7.56 (t, $J$ = 3.5 Hz, 4H), 4.48 (d, $J$ = 2.1 Hz, 2H), 3.09 (d, $J$ = 2.5 Hz, 9H). [13]C NMR (150 MHz, D$_2$O) δ 165.98, 132.80, 130.83, 129.18, 127.42, 69.63, 69.61, 52.40, 52.37, 52.34. Elemental analysis: calcd for C$_{21}$H$_{32}$N$_2$O$_3$: C/N = 9.00 (w/w), found: C/N = 9.24 (w/w).

The thermodynamic effect of counter-cations was assessed by [1]H NMR spectroscopy. A typical sample contained TFP (2 equiv., 0.05 mmol) and the carbonate salt (1 equiv., 0.025 mmol) in 550 μL of CD$_3$CN or a CD$_3$CN/H$_2$O mixture (with H$_2$O added as needed to achieve complete dissolution). After reacting at room temperature for 2 h, the mixture was analyzed directly without purification.

The kinetic effect of counter-cations was studied by UV-Vis spectroscopy. A quartz cuvette was charged with TFP (2 equiv., 0.345 μmol) and the carbonate salt (1 equiv., 0.173 μmol) in 3 mL of CH$_3$CN or a CH$_3$CN/H$_2$O mixture. The decrease in the characteristic TFP absorption at 256 nm was monitored in real time. The conversion ($R\%$) was calculated as $R\% = [(A_0 - A_t) / A_0] \times Y_{final}$, where $A_0$ and $A_t$ are the initial and time-dependent absorbance values, respectively, and $Y_{final}$ is the final yield determined from [1]H NMR.

## Model reaction for the catalyst-free exchange reaction

The exchange reaction was investigated using the preformed $CO_3^{2-}$-bridged adduct (TFPCTFP) and FTFP as model compounds. In a typical procedure, TFPCTFP (22.3 mg, 0.025 mmol, 1 equiv.) and FTFP (9.6 mg, 0.05 mmol, 2 equiv.) were dissolved in 0.6 mL CD$_3$CN or CH$_3$CN. After reacting at room temperature for 10 min, the mixture was directly

analyzed by [1]H and [19]F NMR spectroscopy, as well as mass spectrometry.

## Model reaction for the acid-induced dissociation of $CO_3^{2-}$-bridged adducts

The acid-induced dissociation was investigated using TFPCTFP as the model compound. TFPCTFP (41 mg) was dissolved in 0.5 mL CD$_3$CN to give an 82 mg·mL$^{-1}$ solution. Upon addition of aqueous H$_2$SO$_4$ (0.5 M, 0.5 mL) to adjust the final pH to 4 at room temperature, immediate gas evolution was observed. After stirring for 10 min, the mixture was analyzed directly by [1]H and [19]F NMR spectroscopy. To confirm that the evolved gas was $CO_2$, a gas-capture setup was employed (Supplementary Fig. 15). Solid TFPCTFP (1.23 g, 1.38 mmol) in a Schlenk tube was connected to a second tube containing clear aqueous Ca(OH)$_2$. Upon addition of aqueous H$_2$SO$_4$ (0.5 M, 5.0 mL), vigorous bubbling occurred in the Ca(OH)$_2$ solution, which gradually turned opaque, confirming $CO_2$ release.

## Synthesis of monomer VTFP

A mixture containing 4-bromo-α,α,α-trifluoroacetophenone (15.18 g, 60 mmol, 1 equiv.), K$_2$CO$_3$ (16.70 g, 121 mmol, 2 equiv.), Pd(PPh$_3$)$_4$ (0.72 g, 0.62 mmol, 0.01 equiv.) and pinacol vinylboronate (11.09 g, 72 mmol, 1.2 equiv.) in dioxane/H$_2$O (100/20 mL) were stirred under N$_2$ at 100 °C for 24 hours. After filtration to remove insoluble solids and distillation of dioxane under reduced pressure, the residue was extracted with ethyl acetate (3×), dried over MgSO$_4$, concentrated, and purified by flash chromatography on silica gel to afford VTFP as a colorless oil (9.60 g, 80% yield). [1]H NMR (600 MHz, CDCl$_3$) δ 8.04 (d, $J$ = 12.6 Hz, 2H), 7.56 (d, $J$ = 12.6 Hz, 2H), 6.78 (dd, $J$ = 16.2, 10.2 Hz, 1H), 5.96 (d, $J$ = 26.4 Hz, 1H), 5.51 (d, $J$ = 16.2 Hz, 1H). m/z calcd for C$_{10}$H$_7$F$_3$O, 200.05 [M], Found 200.040[M].

## Synthesis of P(VTFP-MA)-20

VTFP (3.00 g, 15 mmol, 1 equiv.), MA (25.80 g, 300 mmol, 20 equiv.), and AIBN (129 mg, 0.7875 mmol, 1/400 equiv.) were dissolved in 1,4-dioxane (30 mL). After degassing, the mixture was heated at 65 °C for 24 hours. Precipitation into MeOH, filtration, and drying afforded P(VTFP-MA)-20 as white solids (25.50 g, 89%). [1]H NMR (600 MHz, CD$_3$CN) δ 7.90 (d, $J$ = 108 Hz, 2H), 7.28 (d, $J$ = 114 Hz, 2H), 3.60 (s, 63H). Mn = 48.1 KDa, Mw = 73.7 KDa, Đ = 1.53.

## Synthesis of CANs

CANs were synthesized in a one-pot, two-step procedure directly from ambient air. Taking $CO_2$-derived CAN as an example, ambient air was bubbled at ~300 mL·min$^{-1}$ into a solution of TBAH (86 mg, 0.34 mmol, 20 equiv.) in CH$_3$CN/H$_2$O (29:1 v/v, 3 mL) under ambient conditions until the pH reached ~7.5. Subsequently, a solution of P(VTFP-MA)-20 (800 mg, 0.0166 mmol, 0.049 equiv.) in CH$_3$CN (1 mL) was added. Stirring the mixture at room temperature for 40 minutes afforded a cross-linked gel, which was then dried under vacuum for 12 h to yield light-yellow CAN.

## Evaluation of ambient air $CO_2$ capture performance

Ambient air (~420 ppm $CO_2$) was bubbled at 300 mL·min$^{-1}$ through a solution of TBAH (86 mg, 0.34 mmol, 1 equiv.) in CH$_3$CN/H$_2$O (29:1 v/v, 3 mL) at room temperature until the pH reached ~7.5—a value corresponding to that of a pure TBAC solution at the same concentration. The effluent $CO_2$ concentration was monitored in real time using a gas detector (CMF9002, Consensic, Inc.). Gravimetric analysis showed the as-synthesized product to be 94.1% TBAC, containing ~5.9% TBAHC. The product was characterized by [1]H NMR, [13]C NMR, and IR spectroscopy. [1]H NMR (600 MHz, D$_2$O) δ 3.19 (m, 16H), 1.65 (p, $J$ = 7.2, 8.4 Hz, 16H), 1.35 (h, $J$ = 7.2 Hz, 16H), 0.95 (t, 7.2 Hz, 24H). [13]C NMR (150 MHz, CDCl$_3$): δ 160.3, 58.6, 23.9, 19.6, 13.7. IR (ATR): ν 2961, 1614, 1489, 1363, 966, 828, 757, 677, 468 cm$^{-1}$.

### Effect of trace $HCO_3^-$ on the network

To evaluate the influence of trace $HCO_3^-$ impurity, a control CAN was synthesized using purified TBAC. A solution of P(VTFP-MA)-20 (800 mg, 0.0166 mmol, 0.049 equiv.) in $CH_3CN$ (1 mL) was treated with TBAC (90.7 mg, 0.167 mmol, 10 equiv.) and stirred for 30 min at room temperature. After solvent removal, the product was then hot-pressed into rectangular specimens, followed by DSC and tensile testing of the control network against the $CO_2$-derived CAN (Supplementary Fig. 27).

A small-molecule model reaction was further performed to probe the impact at the molecular level. A stoichiometric mixture of TFP (8.7 mg, 0.05 mmol, 1 equiv.) and TBAHC (15.2 mg, 0.05 mmol, 1 equiv.) was stirred in $CD_3CN$ (0.6 mL) at room temperature for 24 hours. The resulting mixture was directly analyzed by $^1H$ and $^{19}F$ NMR spectroscopy (Supplementary Fig. 28).

### Molecular dynamics (MD) simulations

MD simulations were performed using the Large-scale Atomic/Molecular Massively Parallel Simulator (LAMMPS) package[53]. The General Amber Force Field (GAFF2)[54] was employed to describe intramolecular interactions. For intermolecular interactions, the Lennard-Jones potential with a cutoff of 1.2 nm and Coulombic interactions with particle-particle particle-mesh (PPPM) treatment for long-range electrostatics were applied. Five systems were constructed, each containing 50 anionic polymer chains and 100 counter-cations—including $TBA^+$, $TPA^+$, $TEA^+$, $TMA^+$, and $BnMA^+$—within a cubic simulation box of 6.8 nm under periodic boundary conditions using Packmol[55]. Additionally, a control system was built, in which the plasticizing $TBA^+$ cation was replaced by $H^+$, to elucidate the plasticizing role of the cationic structures, with charge neutrality maintained in all cases. Each system underwent energy minimization, followed by a 2 ns equilibration in the isothermal-isobaric (NPT) ensemble at 300 K and 1 atm, controlled by a Nosé-Hoover thermostat with a time step of 1 fs. After equilibration, uniaxial tensile deformation was simulated by stretching each system to 400% of its original length over a period of 10 ns at 300 K and 1 atm.

### Environmental durability assessment

The environmental durability of the $CO_2$-derived CAN was assessed through a series of accelerated aging tests and an outdoor exposure study. The accelerated aging tests included: (1) UV stability: samples were exposed to 365 nm UV radiation (10W) for 5, 7, and 14 days; (2) Humidity stability: samples were conditioned at 25 °C under 15%, 30%, and 45% RH for 24 h; (3) Thermal cycling: samples underwent 3, 6, and 9 complete cycles, with each cycle consisting of 30 min at −20 °C and 30 min at 110 °C. Static outdoor exposure trials were carried out for 4, 6, and 8 weeks under ambient conditions. After each test, samples were characterized for changes in mechanical properties, appearance, and chemical structure.

### Self-healing

Rectangular specimens of the $CO_2$-derived CAN (30 mm × 10 mm × 0.5 mm) were prepared by hot pressing. For visual tracking, a trace amount of rhodamine B was doped into one set of specimens. Each sample was then cut transversely with a blade. The fractured surfaces were placed in contact and healed under controlled pressure and temperature. Standard healing was conducted at 0.5 MPa and 80 °C, whereas low-temperature healing was performed at 40 °C (2 MPa, 24 h) and at room temperature (25 °C, 2 MPa, 24 h). The repaired samples were then evaluated by uniaxial tensile testing.

### Reprocessability

The reprocessability of the $CO_2$-derived CAN was evaluated using compression molding and injection molding. Compression molding was performed by cutting the CAN films into fragments, placing them between steel plates with PTFE films as release layers, and hot-pressing at 90 °C under 6 MPa for 10 min. Injection molding was carried out using a WZS1D micro injection molding machine (Shanghai Xinshuo Precision Machinery Co., Ltd.) at a constant temperature of 100 °C.

### Chemical Recycling

$CO_2$-derived CAN (1.00 g) was fragmented and treated with $H_2SO_4$/$CH_3CN$ solutions (0.2 M or 0.5 M) at room temperature. After complete degradation, P(VTFP-MA)-20 was recycled by liquid-liquid extraction with ethyl acetate and water, while the aqueous phase was concentrated and treated with basic anion-exchange resin to regenerate TBAH. The recycled P(VTFP-MA)-20 and TBAH were reused to synthesize $CO_2$-derived CAN.

To assess real-world applicability, selective recyclability was demonstrated in two scenarios: (1) From mixed plastic waste: A $CO_2$-derived CAN film (150 mg) was combined with polyvinyl chloride (130 mg), polypropylene (106 mg), and polyethylene (118 mg) in 0.5 M $H_2SO_4$/$CH_3CN$ (18 mL). After reacting for 20 min at room temperature, the $CO_2$-derived CAN was completely degraded, while the other plastics remained intact. Following filtration, P(VTFP-MA)-20 and TBAH were recovered as described above. (2) From a carbon-fiber-reinforced polymer (CFRP) composite: A CFRP composite with the $CO_2$-derived CAN as the matrix (364 mg) was treated with 0.5 M $H_2SO_4$/$CH_3CN$ (8 mL) at room temperature. After reacting for 30 min, the CAN matrix degraded completely, leaving the carbon-fiber fabric undamaged. After filtration, P(VTFP-MA)-20 and TBAH were similarly recovered.

### Upcycling

The $CO_2$-derived CAN and the CAN derived from P(VTFP-MA)-1.3 were individually acid-degraded to their respective linear polymers, P(VTFP-MA)-20 and P(VTFP-MA)-1.3. These polymers (P(VTFP-MA)-20: 500 mg, 0.0104 mmol, 1 equiv.; P(VTFP-MA)-1.3: 500 mg, 0.0113 mmol, 1.09 equiv.) were blended in 4 mL of $CH_3CN$ and recross-linked with atmospheric $CO_2$-derived TBAC (406 mg, 0.745 mmol, 71.6 equiv.). After reacting for 40 min at room temperature, the resulting gel was dried under vacuum for 12 h to afford the upcycled hybrid CAN.

### Characterizations

$^1H$, $^{19}F$, and $^{13}C$ spectra were acquired at 298 K on a Bruker Avance III 600 MHz spectrometer using $CDCl_3$, $CD_3CN$, or $D_2O$ as deuterated solvents and referenced to tetramethylsilane (TMS) or residual solvent peaks. FTIR spectra were collected on a PerkinElmer Spectrum Two spectrometer equipped with a diamond attenuated total reflectance (ATR) accessory, with background subtraction for baseline correction. For in situ variable-temperature FTIR analysis, the sample was heated from 25 to 120 °C at a constant rate of 5 °C·min$^{-1}$. UV-Vis absorption spectra were recorded using an Agilent Cary 60 spectrophotometer in quartz cuvettes (0.1 cm path length) at 25 °C. GPC analysis was conducted on a Shimadzu DAWN/LC-20A system with a refractive index detector (RID), employing THF as the eluent at 1.0 mL·min$^{-1}$ and calibrated against polystyrene standards. Elemental analysis was performed to confirm carbonate purity. The pH of solutions was measured using precision pH test strips. The gel fraction and swelling ratio of the CANs were determined by immersing ~230 mg of sample in 1.2 mL of solvent for 12 h at room temperature. The swelling ratio and gel fraction were then calculated as ($m_1$- $m_0$)/$m_1$× 100% and $m_2$/$m_0$ × 100%, respectively, where $m_0$, $m_1$, and $m_2$ are the initial, swollen, and dried sample masses. $T_g$ were determined using a Netzsch DSC 214 instrument under nitrogen atmosphere with heating/cooling rates of 10 °C·min$^{-1}$. Thermal stability was evaluated via TGA using a PerkinElmer TGA 8000 under air atmosphere, heating from 30 °C to 500 °C at 5 °C·min$^{-1}$. Tensile testing was performed on an MTS E44 universal testing machine at room temperature using rectangular samples (30 mm × 10 mm × 0.5 mm) prepared by hot-pressing and precision cutting. At least three replicates were tested for each condition.

Young's modulus was derived from the initial linear region of stress-strain curves, and toughness was calculated as the area under the curve. DMA was carried out on a PerkinElmer DMA 8000 in tension mode at a heating rate of 2 °C·min$^{-1}$ and a fixed frequency of 1 Hz. Rheological behavior was characterized using a TA Instruments DHR-1 rotary rheometer with 25 mm-diameter, 1 mm-thick compression-molded discs. Strain-sweep tests were first conducted at 1 Hz and 60 °C to determine the linear viscoelastic regime. Stress relaxation experiments were then performed at 60 °C under 1% strain. The $\tau$ was defined as the time required for $G'$ to decay to 1/e (-36.7%) of its initial value. SAOS tests were employed from 60 to 100 °C at a fixed strain amplitude of 1%. The master curves were constructed by applying the TTS principle to the SAOS data at a reference temperature of 60 °C. Temperature-sweep tests were conducted at a fixed strain of 1% and a frequency of 1 Hz.

## Data availability

All data generated in this study have been deposited in the Figshare repository under accession code (https://doi.org/10.6084/m9.figshare.30476813). All data are also available from the corresponding author upon request. Source data are provided with this paper.

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

## Acknowledgements

This work was financially supported by the National Natural Science Foundation of China (No. 52573125, L.C.), the Natural Science Foundation of Chongqing (CSTB2025NSCQ-GPX0539, L.C.), the Central University Research Funding Talent Introduction Program (No. SWU-KR23005, L.C.; No. SWU-KQ23015, C. L.), the Chongqing Municipal Full-time Postdoctoral Fellowship Retention (Relocation) Program (No. 7820101128, L.C.), the National Natural Science Foundation of China (No. 22305194, C. L.) and the Opening Project of the State Key Laboratory of Polymer Molecular Engineering (Fudan University) (No. K2024-06, L.C.). The authors gratefully acknowledge the help of the Analytical & Testing Center of Southwest University.

## Author contributions

L.C. conceived the concept and supervised the project. X.Z. and S.Z. performed the majority of experiments. H.L. collected GPC data. H.L. and L.C. jointly designed experiments, analyzed data, and drafted the manuscript. C.L. participated in the discussion. X.Z., S.Z., and H.L. contributed equally to this work.

## Competing interests

The authors declare no competing interests.
