## [Transparent Peer Review file · Nature Communications]

Upcycling of Atmospheric CO₂ to Self-Healing Recyclable Polymers under Ambient Conditions

Corresponding Author: Professor Liang Chen

Version 0:

Reviewer comments:

Reviewer #1

(Remarks to the Author)

The manuscript entitled "Catalyst-Free Upcycling of Air-Captured CO₂ to Self-Healing Recyclable Polymers under Ambient Conditions" presents a groundbreaking approach to addressing the sustainability challenges of polymer production by integrating atmospheric CO₂ valorization with dynamic covalent chemistry. The authors demonstrate a strategy for synthesizing mechanically robust, self-healing, and recyclable polymers under ambient conditions, which holds significant potential for advancing circular materials science.

While the study is innovative and methodologically rigorous, I do not think the manuscript satisfy the high standard of Nature Communications in current form. Below are detailed comments and recommendations:

1. One of the major concerns is that in the introduction part, too many selling points are presented while it is not clearly what is the major pointing of the manuscript. For example, the authors claimed that current approach overcome the high barrier of CO₂ while it is mainly for the reduction of CO₂ to other useful molecules instead of acid-base reaction as demonstrated in current manuscript.

2. The authors highlight CO₂ capture as one of the central focuses of their work: (1) however, in traditional crosslinked polymer systems, the mass proportion of crosslinking agents typically remains extremely low. To establish the actual CO₂ capture performance of their system, the authors should provide detailed experimental data demonstrating the specific capacity, explicitly reporting the mass ratio of absorbed CO₂ to the unit mass of the synthesized polymer materials. (2) Environmental Impact Assessment: Quantify the carbon footprint reduction achieved by this method compared to conventional polymer production (e.g., kg CO₂ eq. per kg polymer).

3. The authors have conducted extensive work on material recycling based on dynamic bonds; however, their presentation primarily relies on schematic diagrams of the materials and their performance depictions (Figures 4 and 5), with a notable lack of physical characterization tests to support the material analysis. It is recommended that they incorporate techniques such as DMA, advanced rheological testing (master curve), or in-situ infrared spectroscopy to validate the self-healing and reprocessing capabilities of the newly formed bonds within the polymer system.

4. Counter-Cation Plasticization Mechanism: While the role of counter-cations in enhancing ductility is proposed, molecular dynamics simulations or variable-temperature NMR could elucidate how cation size/mobility influences chain dynamics and mechanical performance.

5. Durability Under Real-World Conditions: Assess the long-term stability of the polymers under environmental stressors (e.g., UV exposure, humidity, thermal cycling). For instance, do the materials retain self-healing and mechanical properties after prolonged outdoor exposure?

6. The thermogravimetric analysis (TGA) data for the linear polymer P(VTFP-MA) are lacking.

7. Supplementary Scheme for Reaction Pathways: While Figure 1 provides a conceptual overview, a detailed reaction scheme illustrating the stepwise conversion of CO₂ to CO₃²⁻ intermediates, dynamic bond formation, and network architecture would enhance mechanistic understanding. Include molecular-level diagrams of bond exchange and acid-

triggered dissociation processes.

8. Performance Benchmarking: Include a comparative table evaluating key metrics (e.g., energy input, CO₂ conversion efficiency, mechanical properties, recyclability) against state-of-the-art CO₂-derived polymers (e.g., polycarbonates, polyurethanes) and dynamic networks (e.g., CANs). Highlight advantages such as ambient synthesis conditions and catalyst-free reprocessing.

9. Additionally, there are a few problems that should be dealt with before published in any journal. For example:

- (1) The manuscript contains numerous figures lacking error bars, including Figure 5C and the majority of figures in SI.
- (2) The font sizes in Figure 5C are inconsistent and need to be standardized.
- (3) The manuscript contains several basic errors, such as "Supplementary Figs. 1" on page 2 (likely a typo where the singular "Fig." is needed), the misspelled term "waveunmber" in Figure 3C, and "the" in "the demonstrated....." on page 4.
- (4) The layout of Figure 2 should be more neatly arranged.
- (5) The legend order in the figures should be aligned as closely as possible with the data presentation order, such as in Figure 3D.

Reviewer #2

(Remarks to the Author)

In this work, the authors introduced TFP, which is reactive with CO₂ and produces dynamic crosslinking agents, into a polymer. The resultant polymer was mechanically tough and thermoplastic. Due to the high reactivity of TFP, the polymer directly captured CO₂ even from air. Their findings are interesting, and the results of this work should attract the interest of many readers. I suggest the following comments to improve the results.

- 1) I think that alkylammonium hydroxide is acting as a catalyst. In my opinion, "catalyst-free" is not a fact. Would you consider softening the expression in the title?
- 2) As the authors stated in the Introduction, CO₂ consumption is an important target of this study. However, the amount of CO₂ consumed is not addressed. For example, it would be helpful to indicate the amount of CO₂ consumed per gram of polymer, and to compare this with other CO₂-derived polymers such as polyurethanes.
- 3) P. 3, Fig. 3b: In addition to DSC measurements, I recommend performing a temperature sweep using dynamic mechanical analysis (DMA). The DMA results may exhibit a plateau above the T_g for the CO₂-derived CAN, while almost no plateau would be observed for P(VTFP-MA)-20. The storage modulus in the plateau region is related to the effective crosslinking density of the CO₂-derived CAN. It is also possible that a relaxation mode associated with network exchange might be detected above the T_g.
- 4) P. 3, Fig. 3e: The CO₂-derived CAN exhibits an exceptionally high elongation at break of 958%, despite having a crosslinking density as high as 5%, assuming that all TFP sites have reacted. While the authors discuss the influence of counter cations, this reviewer believes that the high elongation may result from the breaking of CO₂-based crosslinks during sample stretching. For example, cyclic tensile testing could be employed to evaluate the plastic deformability of the network; therefore, I recommend performing such tests.
- 5) P. 3, Fig. 3g: The authors indicate that the mechanical properties of the CO₂-derived CANs decrease with increasing alkyl chain length of the alkylammonium hydroxide. Please describe the cause of this result.
- 6) P. 3, Fig. 3h: The authors proposed that the mechanical properties of CO₂-derived CANs are widely tunable, ranging from flexible elastomers to rigid plastics. However, this tunability is mainly due to the characteristics of the polymers used for copolymerization. Therefore, I believe that the advantage of the CO₂-derived CAN may be diminished when rigid EMA and MMA are used. Please compare the mechanical properties before and after CO₂ crosslinking for copolymers with BuMA, EMA, and MMA. If any advantages of CO₂ crosslinking are observed, they should be clearly described.
- 7) P. 4, Fig. 4: Regarding the experiment shown in Fig. 4, there is no detailed description of the CO₂-derived CAN. Please include an explanation of the prepolymer. Is the prepolymer P(VTFP-MA)-20?
- 8) P. 4, Fig. c and f: The authors conducted self-healing tests at 80°C; however, they also demonstrated that melt processing is possible at a slightly higher temperature of 90°C. In such a case, this reviewer believes that the observed behavior may be attributed to thermoplasticity rather than true self-healing. How would the self-healing performance be affected if the tests were conducted at a lower temperature where melt processing is not possible, such as room temperature?

Reviewer #3

(Remarks to the Author)

Comments for manuscript "Catalyst-Free Upcycling of Air-Captured CO₂ to Self-Healing Recyclable Polymers under Ambient Conditions" Nature Sustainability.

The manuscript submitted by Prof. Liang Chen and co-workers deals with the synthesis of covalent adaptable networks using ambient CO₂ as a crosslinker which enable the formation of exchangeable bonds. The manuscript is well-written, the molecular study and the effect of the counter ion on the hemiacetal-like moiety formation is highly interesting. The network characterization and material properties are well performed showing excellent mechanical properties and good (re)processability. The manuscript deserves publication after considering the following points especially further studies on rheological characterizations should be performed prior publication.

- 1) Network design. Can the authors discuss the choice of (co)polymer backbone structure? Why they choose to copolymerize MMA with functionalized styrene and not a fully styrenic polymer backbone?
- 2) Swelling tests in good solvents (e.g. THF or CHCl₃) should be performed for each materials. My issue is that the crosslinked nature has not be evidenced either by gel content measurement or DMA.

3) Rheology experiments to be performed. Stress relaxation experiments presented by the authored showed fast relaxation (between 10 to 100s). Small amplitude oscillatory shear experiments are generally preferred to characterize rapid relaxation (see e.g. ACS Polym. Au 2025, 5, 3, 214–240). It will also confirm the nature of the exchange mechanism (associative or dissociative) by monitoring the evolution of the storage modulus with increasing temperature. In addition, we generally performed stress relaxation experiments at $T > T_g + 50^\circ\text{C}$ to separate the segmental relaxation induce by the glassy to rubbery transition and the relaxation induce by the exchange reaction. Non-normalized data should be also added in the supplementary material.

4) In addition, creep experiments especially at room temperature should be performed to determine if the material mechanical properties are stable under ambient conditions. DMA analysis to further confirm the CAN features (presence of a rubbery after T_g).

Version 1:

Reviewer comments:

Reviewer #1

(Remarks to the Author)

The manuscript entitled "Upcycling of Atmospheric CO₂ to Self-Healing Recyclable Polymers under Ambient Conditions" describes a strategy for producing cross-linked polymers with self-healing and closed-loop recyclability directly from atmospheric CO₂ under catalyst-free conditions, presenting a valuable pathway toward green and reversible crosslinking in polymer science. This study leverages the rapid absorption of atmospheric CO₂ by organic bases such as TBAC, converting it into CO₃²⁻ which enable the dynamic reversible cross-linking of P(VTFP-MA)-20. The resulting covalent network exhibits excellent self-healing properties, recyclability, and favorable mechanical performance.

Overall, the referee believes that the manuscript deserves publication in Nature Communications after the identified issues are addressed.

Here are the minor comments the reviewer suggests:

1. Regarding Fig. 3c, it would be more informative to plot the pH profile of the TBAH solution over time. The current format, which only labels the initial and final pH values, could be misleading to readers.
2. While the model reactions effectively prove CO₃²⁻ reacts with TFP, the study does not address whether HCO₃⁻ could also participate in or interfere with this reaction. During the cross-linking of the linear polymer, ambient air was bubbled through the TBAC solution. This process may inevitably produce HCO₃⁻. It is important to clarify whether the presence of HCO₃⁻ competes with CO₃²⁻ and adversely affects the efficiency of the cross-linking reaction.
3. Referee believes that the statement "the urgent need for novel catalyst-free dynamic chemistries" in the introduction part is inaccurate, since many dynamic covalent bonds can undergo efficient exchange under thermal activation without catalyst. The reviewer recommends that the authors revise this statement to ensure accurate expression.
4. The introduction on the up-to-date progress in CO₂ utilization could be strengthened. It would be beneficial to cite recent achievements in this field (see e.g. Nat. Commun., 15, 6605 (2024)) to provide a more comprehensive background of the present work.
5. Additionally, there are a few problems that should be dealt with before published. For example:
(1) The letter labels (j, k, l) in the caption of Fig. 3 should be bolded and must conform to standard typesetting conventions.
(2) In Fig. 5b, a space should be inserted between the axis label name and the following parentheses.

Reviewer #2

(Remarks to the Author)

The authors have conducted additional experiments and have appropriately addressed the reviewer's questions. I recommend accepting this paper in its current version.

Reviewer #3

(Remarks to the Author)

The revised version of the manuscript entitled "Upcycling of Atmospheric CO₂ to Self-Healing Recyclable Polymers under Ambient Conditions" includes all the suggested modifications especially about the rheological behavior of these original materials. I therefore recommend publication of this manuscript.

Version 2:

Reviewer comments:

Reviewer #1

(Remarks to the Author)

I think the authors did a pretty good work on addressing the comments from reviewers and I would recommend the acceptance of the manuscript.

Point-by-Point Response to Reviewers' Comments

Reviewer #1:

The manuscript entitled "Catalyst-Free Upcycling of Air-Captured CO₂ to Self-Healing Recyclable Polymers under Ambient Conditions" presents a groundbreaking approach to addressing the sustainability challenges of polymer production by integrating atmospheric CO₂ valorization with dynamic covalent chemistry. The authors demonstrate a strategy for synthesizing mechanically robust, self-healing, and recyclable polymers under ambient conditions, which holds significant potential for advancing circular materials science.

While the study is innovative and methodologically rigorous, I do not think the manuscript satisfy the high standard of Nature Communications in current form. Below are detailed comments and recommendations:

Response: We sincerely thank you for your time and insightful feedback on our manuscript, especially regarding CO₂ capture performance assessment and the need to better articulate our core innovation. We have carefully considered all your comments and implemented substantive, targeted revisions to address them. Detailed point-by-point responses are provided below for your review.

Comment 1: One of the major concerns is that in the introduction part, too many selling points are presented while it is not clearly what is the major pointing of the manuscript. For example, the authors claimed that current approach overcome the high barrier of CO₂ while it is mainly for the reduction of CO₂ to other useful molecules instead of acid-base reaction as demonstrated in current manuscript.

Response: Thank you for your valuable feedback regarding the lack of focus in the original introduction and the specific point concerning the claim of "overcoming the high barrier of CO₂".

We acknowledge that the original introduction presented too many points without sufficiently highlighting the central innovation. The core advance of our work is the development of a novel dynamic covalent chemistry (DCC) enabling catalyst-free conversion of atmospheric CO₂ into high-performance recyclable polymers under ambient conditions. To sharpen this focus, we have implemented significant revisions: *(1) Streamlined discussion of CAN limitations (Paragraph 1)*: We now concentrate specifically on challenges most relevant to our solution—harsh reprocessing conditions, limited recyclability, and mechanical property trade-offs—avoiding excessive detail while clearly establishing the need for novel dynamic covalent bonds. *(2) Restructured positioning of DACU (Paragraph 2)*: We explicitly position direct air capture and utilization (DACU) as the paramount unsolved challenge for CO₂-derived polymers. The revised text now highlights the fundamental barriers our work addresses: directly converting ultradilute atmospheric CO₂ (~400 ppm) into functional polymers under ambient, catalyst-free conditions. *(3) Refocused materials fabrication narrative*: To better emphasize this central innovation, we have removed the content related to CANs derived from pure CO₂ in the 'Synthesis and Characterization of CANs' section, refocusing all the polymer networks synthesis narrative on their direct utilization and conversion from ambient air.

Regarding the specific point about CO₂ activation, we appreciate this opportunity for clarification. We agree that catalytic CO₂ reduction pathways represent a significant frontier in carbon utilization; however, these predominantly target small molecule synthesis (e.g., CO₂-to-fuels) rather than polymers. Well-established CO₂-to-polymer methodologies—such as CO₂/epoxide copolymerization, CO₂/olefin cyclization, CO₂/alkyne cyclization, and CO₂/Frustrated Lewis pairs

(FLP) coupling—all operate through non-reductive mechanisms (*Chem. Soc. Rev.* **2019**, 48, 4466-4514; *Cell Rep. Phys. Sci.* **2022**, 3, 100719). Distinct from these approaches, our work presents the first example of conversion of CO₂ into CO₃²⁻ intermediates for polymer synthesis. This strategy delivers three key advantages: (i) direct utilization of ultra-dilute atmospheric CO₂ (~400 ppm), (ii) catalyst-free operation under mild conditions (room temperature, atmospheric pressure), and (iii) the generation of polymers with excellent self-healing and recyclability. We recognize that the original manuscript mention of “CO₂'s fully oxidized state” (Paragraph 3) could have been construed as implying that conventional CO₂-to-polymer methods operate via reduction mechanisms, which was not our intention. To eliminate ambiguity and accurately reflect our approach, we have revised the opening of Paragraph 3 to: “*The fundamental challenge in converting CO₂ arises from its inherent stability and inertness, yet, ...*”.

We believe these revisions provide a clearer, more focused, and accurate introduction that properly frames our research on atmospheric CO₂ conversion to recyclable polymers. Thank you again for this helpful critique.

Comment 2: The authors highlight CO₂ capture as one of the central focuses of their work: (1) however, in traditional crosslinked polymer systems, the mass proportion of crosslinking agents typically remains extremely low. To establish the actual CO₂ capture performance of their system, the authors should provide detailed experimental data demonstrating the specific capacity, explicitly reporting the mass ratio of absorbed CO₂ to the unit mass of the synthesized polymer materials. (2) Environmental Impact Assessment: Quantify the carbon footprint reduction achieved by this method compared to conventional polymer production (e.g., kg CO₂ eq. per kg polymer).

Response: Thank you for this insightful comment. We agree that the mass fraction of crosslinkers in conventional polymer networks is generally low. Our initial experiments indeed explored TFP-based small molecules to maximize the CO₂ content; however, these attempts encountered synthetic challenges and yielded materials with unsatisfactory mechanical properties. We therefore selected the linear polymer (P(VTFP-MA)-20) as our optimal strategy, as it enables both a simplified synthesis and excellent material properties. While the mass of CO₂ incorporated per unit polymer mass is currently limited, we should note that our work establishes a distinct and valuable paradigm, characterized by: (i) direct utilization of CO₂ from ambient air (~400 ppm), which remains unattainable by existing CO₂-to-polymer methods requiring concentrated pure CO₂ sources; (ii) catalyst-free CO₂ fixation under exceptionally mild conditions (room temperature, atmospheric pressure), unlike literature approaches that often involve complex catalysts, high pressures, and elevated temperatures; and (iii) production of polymers with excellent intrinsic self-healing and recyclability. Moreover, the current CO₂ uptake has not yet reached the theoretical maximum, and this theoretical value can be further increased by optimizing the CO₃²⁻-bonding groups via lower-molecular-weight alternatives, such as trifluoromethyl alkyl ketones.

To address your comment directly, we have performed the following additional experiments and analyses:

(1) *CO₂ Capture Capacity.* To evaluate CO₂ capture performance, we first conducted corresponding CO₂ capture experiments by bubbling ambient air into an acetonitrile (CH₃CN)/H₂O solution of tetrabutylammonium hydroxide (TBAH) under ambient conditions. As shown in **Fig. R1**, the CO₂ capture efficiency—defined as the mass ratio of retained CO₂ in the system to the total CO₂ supplied from ambient air—was determined to be 80.0%, demonstrating highly efficient

fixation even under dilute CO₂ conditions. Furthermore, the CO₂ uptake capacity of the CANs was calculated to be 1.4-5.4 wt% (0.014-0.054 kg CO₂ per kg polymer). This value is lower than that of conventional CO₂-derived polycarbonates and polyurethanes, and is comparable to that reported for frustrated Lewis pair (FLP) systems (see Supplementary Tables 1-2 in the Supporting Information).

(2) *Environmental Impact Assessment.* Although the mass of incorporated CO₂ remains limited, the environmental carbon footprint of the production process is highly favorable. The catalyst-free synthesis, which operates under ambient conditions and directly utilizes atmospheric CO₂, circumvents the substantial energy consumption and emissions associated with conventional CO₂ capture, concentration, compression, and catalytic conversion processes. Furthermore, the synthesis of the P(VTFP-MA)-20 precursor is straightforward, efficient, and inherently scalable. Our preliminary carbon footprint assessment, conducted per the ISO 14040 series using SimaPro (v9.0) software, indicates that the net carbon footprint of our CANs is comparable to epoxy resin (EP) and polystyrene (PS), but lower than those of polyamide (PA), polycarbonate (PC) and polyurethane (PU) foam (**Fig. R2a**). This advantage is attributed to the direct utilization of atmospheric CO₂—which partially displaces petroleum-based feedstocks—coupled with significantly lower energy consumption during manufacturing (**Fig. R2b**). Moreover, the inherent reprocessability of our CANs further minimizes their lifetime environmental impact by extending their service life over multiple cycles, thereby underscoring their environmental advantage. While the production process itself exhibits a favorable carbon footprint, we acknowledge that the contribution of direct air capture and utilization (DACU) is still limited at the current stage. We are actively pursuing strategies—including optimized network design and improved synthetic routes—to further increase CO₂ uptake and enhance the thermal and mechanical properties. A comprehensive life-cycle assessment (LCA) is also underway to more accurately quantify the full environmental benefits of this approach.

Data on CO₂ capture efficiency, along with the corresponding experimental details and discussion, have been incorporated into the revised manuscript (Page 3). Besides, to provide a balanced perspective, the Conclusions section now explicitly states the current limitation of CO₂ uptake, presents the quantitative data, and outlines our ongoing research direction to enhance CO₂ uptake and improve the thermal and mechanical properties (Page 6).

Fig. R1 (newly added Fig. 3d) | CO₂ concentration in the exhaust stream as function of time during the bubbling of ambient air into the TBAH solution.

Fig. R2 | **a**, Greenhouse gas (GHG) emissions from the production of CANs versus conventional polymers. **b**, Main contributors to the GHG emissions from CANs production.

Comment 3: The authors have conducted extensive work on material recycling based on dynamic bonds; however, their presentation primarily relies on schematic diagrams of the materials and their performance depictions (Figures 4 and 5), with a notable lack of physical characterization tests to support the material analysis. It is recommended that they incorporate techniques such as DMA, advanced rheological testing (master curve), or *in-situ* infrared spectroscopy to validate the self-healing and reprocessing capabilities of the newly formed bonds within the polymer system.

Response: We sincerely thank you for this insightful and constructive comment. In response, we have conducted the following additional experiments:

(1) Dynamic Mechanical Analysis (DMA): As shown in **Fig. R3**, the CO₂-derived CAN exhibits a distinct rubbery plateau in the storage modulus (E') above its T_g , which contrasts strikingly with the linear polymer where E' drops continuously with increasing temperature. This sustained modulus is a key indicator of a dynamic covalent network. The crosslink density (ν) was calculated as $79.1 \text{ mol} \cdot \text{m}^{-3}$ using the equation $\nu = E'_r/[3R(T_g + 50)]$.

(2) Rheological testing: The linear viscoelastic regime (LVR) was first determined through strain sweep tests within a strain amplitude (γ) range of 0.001%-10% (**Fig. R4a**). Subsequently, frequency-sweep small-amplitude oscillatory shear (SAOS) tests were carried out at various temperatures (**Fig. R4b**). The relaxation time (τ) was derived from the crossover frequency (ω_c) where the storage modulus (G') equals the loss modulus (G''), using the equation $\tau = 1/\omega_c$. Arrhenius analysis revealed a linear correlation between $\ln(\tau)$ and $1000/T$ (**Fig. R4c**), confirming the vitrimeric nature of the networks. The activation energy (E_a) for the solid-state bond exchange, calculated from the Arrhenius plot, was 111.9 kJ/mol.

Furthermore, a master curve was constructed by applying the time-temperature superposition (TTS) principle to the SAOS data at a reference temperature of 60 °C. The data exhibited a clear terminal relaxation behavior in the low-frequency region, characterized by a transition from solid-like ($G' > G''$) to liquid-like ($G'' > G'$) response (**Fig. R4d**). The horizontal shift factors (a_T) obtained from the TTS procedure were analyzed using the Arrhenius equation (**Fig. R4e**), yielding an activation energy of 109.3 kJ/mol. This value is consistent with that obtained from the crossover frequency method, thereby validating the reliability of our rheological analysis.

To elucidate the exchange mechanism, temperature-sweep experiments were performed. The results showed a sharp, nearly three-order-of-magnitude decrease in G' , accompanied by a distinct G'/G'' crossover at 85 °C (**Fig. R4f**). This behavior is characteristic of a dissociative exchange mechanism, marked by a sudden loss of network integrity upon heating.

(3) *In situ variable-temperature FTIR*: To further corroborate the dissociative exchange mechanism, we conducted *in situ* variable-temperature FTIR spectroscopy (**Fig. R5**). As the temperature increased from 25 to 120 °C, a gradual intensification of the absorption band at 1730 cm^{-1} was observed, indicating the thermally induced cleavage of the CO_3^{2-} -bridged dynamic covalent bonds and regeneration of the TFP compound. Upon subsequent cooling to room temperature for 24 hours, the original CO_3^{2-} -bridged bonds were restored, confirming the reversible and dissociative nature of the CAN.

Overall, the combination of DMA, advanced rheological analyses, and *in situ* FTIR spectroscopy provides compelling evidence for the dynamic bond exchange in our polymer system. These results collectively validate the self-healing and reprocessing capabilities of the newly formed bonds within the polymer system, directly addressing your insightful comment and significantly strengthening our manuscript. All new data and corresponding discussion have been incorporated into the revised manuscript (Manuscript: Pages 4-5; Supporting Information: Pages 26-27).

Fig. R3 (newly added Fig. 3g) | DMA curves of P(VTFP-MA)-20 and CO_2 -derived CAN.

Fig. R4 (newly added Fig. 4a-c and Supplementary Figs. S42-44) | **a**, Strain sweep curves of CO_2 -derived CAN. **b**, **b-c**, Frequency-sweep rheological measurements at various temperatures (**b**) and the corresponding Arrhenius plot of relaxation time (τ) versus $1000/T$ (**c**). **d-e**, Master curve of CO_2 -derived CAN based on TTS of SAOS experiments at a reference temperature of 60 °C (**d**), and the corresponding Arrhenius plot of horizontal shift factors (a_T) versus $1000/T$ (**e**). **f**, Temperature-dependent rheological behavior.

Fig. R5 (newly added Fig. 4d) | *In situ* variable-temperature FTIR spectra of CO₂-derived CAN.

Comment 4: Counter-Cation Plasticization Mechanism: While the role of counter-cations in enhancing ductility is proposed, molecular dynamics simulations or variable-temperature NMR could elucidate how cation size/mobility influences chain dynamics and mechanical performance.

Response: Thank you for this valuable suggestion. As recommended, we have conducted comprehensive molecular dynamics (MD) simulations to elucidate the plasticization mechanism and to systematically investigate the influence of different counter-cations on the mechanical properties.

First, to directly decipher the plasticizing role of cations, we performed a comparative analysis between the CO₂-derived CAN (with TBA⁺) and a control system (with H⁺ replacing TBA⁺). This comparison revealed key distinctions: (i) a weaker ionic association, as indicated by the significantly right-shifted first peak in the radial distribution function (RDF) between TBA⁺ and the polymeric anions (Fig. R6a); (ii) significantly enhanced cation mobility, with a diffusion coefficient (D) 2.1 times greater than that of the control group (Fig. R6b-c); (iii) a considerably expanded polymer free volume fraction (FFV, Fig. R6d-e), which provides more space for segmental motion; and (iv) dramatically accelerated relaxation dynamics of the polymer chains, as demonstrated by a significantly increased mean-squared displacement (MSD) over time (Fig. R6f-g). These findings collectively demonstrate that the bulky, flexible TBA⁺ cations effectively screen electrostatic interactions, enhance mobility, expand free volume, and lubricate the polymer chains, thereby rationalizing the enhanced ductility and toughness.

Second, building on this mechanistic insight, we further investigated how different counter-cations govern the macroscopic mechanical properties. We conducted a systematic comparison across a series of cations (TBA⁺ → TPA⁺ → TEA⁺ → TMA⁺ → BnMA⁺), which revealed a clear trend: reducing the cation size and increasing its rigidity progressively attenuates the plasticizing effect. This attenuation was quantitatively captured by our MD simulations, which showed a left-shift in the RDF peaks (Fig. R6 a), a concomitant decrease in cation mobility (Fig. R6b-c), and a reduction in polymer free volume (Fig. R6d-e) along the cation series. Consequently, the segmental motion of the polymer chains becomes more restricted (Fig. R6f-g), shifting the macroscopic mechanical behavior from highly ductile to more rigid and strong. This structure-property relationship provides a powerful design principle for tailoring the mechanics of our CANs.

All new data from these MD simulations and the corresponding discussion have been incorporated into the revised manuscript (Manuscript: Page 4; Supporting Information: Pages 18-20).

Fig. R6 (newly added Fig. 3i-l and Supplementary Figs. 32-33) | Molecular dynamics (MD) simulations of the counter-cation plasticization mechanism. **a**, Radial distribution function (RDF) between cations and polymeric anions, indicating ionic association strength. **b-c**, Mean-squared displacement (MSD) of counter-cations (**b**) and their corresponding diffusion coefficients (**c**). **d-e**, Polymer free volume fraction (FFV) of the CANs. **f-g**, MSD of polymer backbone (**f**) and their corresponding diffusion coefficients (**g**).

Comment 5: Durability Under Real-World Conditions: Assess the long-term stability of the polymers under environmental stressors (e.g., UV exposure, humidity, thermal cycling). For instance, do the materials retain self-healing and mechanical properties after prolonged outdoor exposure?

Response: Thank you for your insightful suggestion regarding the environmental durability of our CO₂-derived CANs. In response, we have performed a series of accelerated aging tests to simulate real-world conditions: (1) UV resistance was evaluated by exposing samples to UV radiation (10 W, 365 nm) for 5, 7, and 14 days; (2) Humidity stability was assessed by conditioning samples at 25 °C under 15%, 30% and 45% relative humidity (RH) for 24 hours; (3) Thermal cycling tests were conducted for 3, 6, and 9 cycles (each cycle: 30 minutes at -20 °C and 30 minutes at 110 °C). The comprehensive data, presented in **Figs. R7-9**, show that the CO₂-derived polymers retain their mechanical properties, appearance, and chemical structure, thereby confirming their robust stability.

To further evaluate practical durability, outdoor exposure trials were conducted. Results showed that the networks maintained their mechanical properties, appearance and structural integrity after 4, 6, and 8 weeks of exposure (**Fig. R10a-c**), underscoring their real-world application potential. Furthermore, self-healing experiments were conducted on the samples after 8 weeks of outdoor

exposure. The results showed a high healing efficiency of 98.4%, with nearly complete recovery of mechanical properties (**Fig. R10 d-f**), confirming the retention of functional integrity.

All relevant data, experimental details, and extended discussions have been incorporated into the revised manuscript (Manuscript: Page 4; Supporting Information: Pages 24-25).

Fig. R7 (newly added Supplementary Fig. 38) UV stability of the CO₂-derived CAN. **a**, Stress-strain curves before and after various UV exposure durations. *Inset*: Representative photographs of a sample before and after 14 days of UV exposure. **b**, Young's modulus and toughness from (a). **c**, FTIR spectra before and after various UV exposure durations.

Fig. R8 (newly added Supplementary Fig. 39) Humidity stability of the CO₂-derived CAN. **a**, Stress-strain curves before and after conditioning at various relative humidity (RH) levels for 24 hours. *Inset*: Representative photographs of a sample before and after 24 hours of 45% RH exposure. **b**, Young's modulus and toughness from (a). **c**, FTIR spectra before and after conditioning at various RH levels for 24 hours.

Fig. R9 (newly added Supplementary Fig. 40) Thermal cycling stability of the CO₂-derived CAN. **a**, Stress-strain curves before and after various thermal cycles. *Inset*: Representative photographs of a sample before and after 9 thermal cycles. **b**, Young's modulus and toughness from (a). **c**, FTIR spectra before and after various thermal cycles.

Fig. R10 (newly added Supplementary Fig. 41) Outdoor stability and retention of self-healing properties in the CO₂-derived CAN. **a-c**, Outdoor stability assessment. **a**, Stress-strain curves before and after various outdoor exposure durations. *Inset*: Representative photographs of a sample before and after 8 weeks of outdoor exposure. **b**, Young's moduli and toughness from (a). **c**, FTIR spectra before and after various outdoor exposure durations. **d-f**, Self-healing performance after 8 weeks of outdoor exposure: **(d)** Photographs showing the self-healing process of an 8-week outdoor-exposed sample; **(e)** Comparative stress-strain curves of the original and the healed sample; **(f)** Young's modulus and toughness derived from (e).

Comment 6: The thermogravimetric analysis (TGA) data for the linear polymer P(VTFP-MA) are lacking.

Response: Thank you for raising this point. In response, we conducted TGA analyses on both the linear polymer P(VTFP-MA)-20 and TBAC (**Fig. R11**). The results show that P(VTFP-MA)-20 possesses high thermal stability, with a 5% weight loss temperature ($T_{d, 5\%}$) of 315 °C, whereas TBAC decomposes at a much lower temperature of 173 °C. Although the CO₂-derived CAN shows reduced thermal stability ($T_{d, 5\%} = 252^\circ\text{C}$) compared to linear polymer P(VTFP-MA)-20, its onset decomposition temperature of 220 °C confirms that the formation of the network effectively stabilizes the thermally labile TBAC component. These new data and the corresponding discussion have been added to the Supporting Information (Page 17 in the Supporting Information).

Fig. R11 (replaced with Supplementary Fig. 28) | TGA curves of CO₂-derived CAN, linear polymer P(VTFP-MA) and TBAC under air atmosphere.

Comment 7: Supplementary Scheme for Reaction Pathways: While Figure 1 provides a conceptual overview, a detailed reaction scheme illustrating the stepwise conversion of CO₂ to CO₃²⁻ intermediates, dynamic bond formation, and network architecture would enhance mechanistic understanding. Include molecular-level diagrams of bond exchange and acid-triggered dissociation processes.

Response: We thank the reviewer for this suggestion. As recommended, a detailed scheme illustrating the stepwise CO₂ conversion, dynamic bond formation, network architecture, and molecular-level exchange/dissociation processes has now been added to the revised manuscript as Fig. R12.

Fig. R12 (replaced with Fig. 1) | Methodologies for sustainable polymers. a, Reversible TFP-CO₃²⁻ reaction. **b,** Conversion of atmospheric CO₂ into CO₃²⁻ as intermediates for the catalyst-free synthesis of self-healing recyclable polymers under ambient conditions.

Comment 8: Performance Benchmarking: Include a comparative table evaluating key metrics (e.g., energy input, CO₂ conversion efficiency, mechanical properties, recyclability) against state-of-the-art CO₂-derived polymers (e.g., polycarbonates, polyurethanes) and dynamic networks (e.g., CANs). Highlight advantages such as ambient synthesis conditions and catalyst-free reprocessing.

Response: Thank you for this insightful suggestion. As suggested, we have added two comparative tables (Supplementary Tables 1 and 2) to the revised Supporting Information. These tables benchmark our CAN against state-of-the-art CO₂-derived polymers and dynamic covalent networks across key metrics, including CO₂ source, CO₂ conversion rate, CO₂ content, synthesis condition (catalyst, temperature, and pressure), mechanical properties, recyclability (chemical recyclability, reprocessability), self-healing property. The comparison clearly highlights the unique advantages of our system, such as the direct utilization of ambient CO₂, catalyst-free CO₂ fixation at room temperature and atmospheric pressure, excellent self-healing and recyclability. We are grateful for this valuable input.

Comment 9: Additionally, there are a few problems that should be dealt with before published in any journal. For example:

(1) The manuscript contains numerous figures lacking error bars, including Figure 5C and the majority of figures in SI.

Response: Thank you for pointing this out. Error bars have been added to all relevant quantitative plots.

(2) The font sizes in Figure 5C are inconsistent and need to be standardized.

Response: Font sizes in Fig. 5C have been standardized. Thank you for pointing this out.

(3) The manuscript contains several basic errors, such as "Supplementary Figs. 1" on page 2 (likely a typo where the singular "Fig." is needed), the misspelled term "waveunmber" in Figure 3C, and "the" in "the demonstrated……" on page 4.

Response: We apologize for these oversights. All noted errors have been carefully corrected.

(4) The layout of Figure 2 should be more neatly arranged.

Response: Fig. 2 has been reorganized for better visual clarity. Thank you for this valuable suggestion.

(5) The legend order in the figures should be aligned as closely as possible with the data presentation order, such as in Figure 3D.

Response: Thanks for this comment. The legend order in all figures in the revised manuscript is now aligned with the data presentation.

Reviewer #2:

In this work, the authors introduced TFP, which is reactive with CO₂ and produces dynamic crosslinking agents, into a polymer. The resultant polymer was mechanically tough and thermoplastic. Due to the high reactivity of TFP, the polymer directly captured CO₂ even from air. Their findings are interesting, and the results of this work should attract the interest of many readers. I suggest the following comments to improve the results.

Response: We sincerely appreciate your insightful comments. We have carefully considered all of your questions and have addressed them accordingly. Detailed responses are provided below.

Comment 1: I think that alkylammonium hydroxide is acting as a catalyst. In my opinion, "catalyst-free" is not a fact. Would you consider softening the expression in the title?

Response: We sincerely appreciate your insightful comment. In response, we have revised the title to "*Upcycling of Air-Captured CO₂ into Self-Healing Recyclable Polymers under Ambient Conditions*" by removing the phrase "*Catalyst-Free*".

To provide the rationale for retaining the term "catalyst-free" in the main text, we wish to politely clarify the role of alkylammonium hydroxide. According to the IUPAC definition (<https://doi.org/10.1351/goldbook.C00876>), a catalyst is a substance that "increases the rate of a reaction without modifying the overall standard Gibbs energy change" and is "both a reactant and product of the reaction," meaning it is regenerated unchanged after the reaction. In our system, however, alkylammonium hydroxide is stoichiometrically and irreversibly consumed to form carbonate salts, which subsequently act as crosslinkers in the polymer network formation (Fig. R13). Since it is not regenerated, we classify it as a reactant, not a catalyst.

Therefore, we believe that describing the system as "catalyst-free" in the main text remains appropriate, as it correctly highlights the absence of conventional catalysts for CO₂ activation (such as metal complexes)—a key innovative aspect of our work. We hope this clarification addresses your concern.

Thank you once again for your valuable guidance, which has certainly enhanced the clarity and precision of our manuscript.

Fig. R13 | Schematic illustration of the catalyst-free conversion of CO₂ into polymer network.

Comment 2: As the authors stated in the Introduction, CO₂ consumption is an important target of this study. However, the amount of CO₂ consumed is not addressed. For example, it would be helpful to indicate the amount of CO₂ consumed per gram of polymer, and to compare this with other CO₂-derived polymers such as polyurethanes.

Response: Thank you for this valuable suggestion. In our study, the CO₂-derived polymer networks demonstrate a CO₂ uptake capacity of 1.4-5.4 wt% (0.014-0.054 kg CO₂ per kg polymer). This value is lower than that of conventional CO₂-derived polycarbonates and polyurethanes but is comparable

to values reported for frustrated Lewis pair (FLP) systems. While the mass of CO₂ incorporated per unit polymer mass is currently limited, we should note that our work establishes a distinct and valuable paradigm, characterized by: (i) direct utilization of CO₂ from ambient air (~400 ppm), which remains unattainable by existing CO₂-to-polymer methods requiring concentrated pure CO₂ sources; (ii) catalyst-free CO₂ fixation under exceptionally mild conditions (room temperature, atmospheric pressure), unlike literature approaches that often involve complex catalysts, high pressures, and elevated temperatures; and (iii) the generation of polymers with excellent intrinsic self-healing and recyclability. Moreover, the current CO₂ uptake has not yet reached the theoretical maximum, and this theoretical value can be further increased by optimizing the CO₃²⁻-bonding groups using lower-molecular-weight alternatives, such as trifluoromethyl alkyl ketones.

To directly address this point, we have added two tables (Supplementary Tables 1 and 2) to the revised Supporting Information (Pages 46-47), which provides a systematic comparison of key metrics, including CO₂ source, CO₂ conversion rate, CO₂ content, synthesis condition (catalyst, temperature, and pressure), mechanical properties, recyclability (chemical recyclability, reprocessability), self-healing property, between our work and prior CO₂-derived polymers. We believe this comparison will help readers better contextualize the CO₂ uptake level while appreciating the unique advantages of our methodology.

Furthermore, to ensure a balanced perspective, we have revised the Conclusions section to explicitly state this limitation, present the quantitative data, and outline our ongoing research direction to enhance CO₂ uptake and improve the thermal and mechanical properties of the atmospheric-CO₂-derived polymers (Page 6 in the revised manuscript).

Comment 3: P. 3, Fig. 3b: In addition to DSC measurements, I recommend performing a temperature sweep using dynamic mechanical analysis (DMA). The DMA results may exhibit a plateau above the T_g for the CO₂-derived CAN, while almost no plateau would be observed for P(VTFP-MA)-20. The storage modulus in the plateau region is related to the effective crosslinking density of the CO₂-derived CAN. It is also possible that a relaxation mode associated with network exchange might be detected above the T_g .

Response: Thank you for this valuable suggestion. As recommended, we performed temperature-sweep DMA experiments. The results provide clear evidence of a crosslinked network: the CO₂-derived CAN exhibits a well-defined rubbery plateau in the storage modulus (E') above its T_g , starkly contrasting with the continuous modulus decline of the linear polymer precursor (**Fig. R14**). The crosslink density (ν) was calculated as 79.1 mol m⁻³ from the rubbery plateau modulus (E'_r) using the equation $\nu = E'_r/[3R(T_g + 50)]$. Furthermore, upon further heating beyond the rubbery plateau, a subtle but measurable gradual decrease in E' was observed, indicative of the dissociative CAN.

To further elucidate this dissociative exchange mechanism, we conducted additional temperature-sweep rheological tests and *in situ* variable-temperature FTIR. The rheological tests revealed a sharp, nearly three-order-of-magnitude decrease in the storage modulus (G') alongside a distinct G'/G'' crossover at approximately 85 °C (**Fig. R15a**), consistent with a dissociative exchange mechanism. Moreover, variable-temperature FTIR spectra directly demonstrated the reversible cleavage and reformation of the CO₃²⁻-bridged dynamic bonds during heating and cooling process (**Fig. R15b**).

All new data and corresponding discussions have been incorporated into the revised manuscript (Pages 4-5 in the revised manuscript).

Fig. R14 (newly added Fig. 3f) | DMA curves of P(VTFP-MA)-20 and CO₂-derived CAN.

Fig. R15 (newly added Fig. 4c-d) | a, Temperature-dependent rheological behavior of the CO₂-derived CAN. **b,** *In situ* variable-temperature FTIR spectra of CO₂-derived CAN.

Comment 4: P. 3, Fig. 3e: The CO₂-derived CAN exhibits an exceptionally high elongation at break of 958%, despite having a crosslinking density as high as 5%, assuming that all TFP sites have reacted. While the authors discuss the influence of counter cations, this reviewer believes that the high elongation may result from the breaking of CO₂-based crosslinks during sample stretching. For example, cyclic tensile testing could be employed to evaluate the plastic deformability of the network; therefore, I recommend performing such tests.

Response: Thank you for this insightful comment on the exceptional elongation at break of our CO₂-derived CAN. The suggestion that dynamic bond breakage could contribute prompted us to further investigate its origin.

To clarify the underlying mechanism, we considered two plausible pathways: dynamic breaking of CO₃²⁻ crosslinks (as suggested) and plasticization by counter-cations. To inform our investigation, we first surveyed the literature, which indicates that introducing dynamic covalent crosslinks typically restricts chain mobility and reduces elongation at break compared to linear precursors (*J. Am. Chem. Soc.* **2019**, 141, 16595-16599; *Science* **2017**, 356, 62-65). Conversely, small-molecule plasticizers are known to enhance ductility in CANs by disrupting polymer-polymer interactions (*J. Am. Chem. Soc.* **2018**, 140, 6217-6220; *Int. J. Mol. Sci.* **2024**, 25, 1720). Guided by this understanding, we designed control experiments to explicitly distinguish between these two mechanisms. *(i) Ionic Plasticization in Linear Polymer:* Blending tetrabutylammonium iodide (TBAI) into the linear precursor P(VTFP-MA)-20 significantly increased its elongation at break from 603% to 878%, while concurrently reducing its strength and stiffness (Fig. R16a). This result confirms the potent plasticizing effect of ionic species. *(ii) Substitution with a Dynamic Covalent Crosslinker:* We replaced the tetrabutylammonium carbonate (TBAC) linkages with 1,3-diaminopropane crosslinks to form a dynamic network, defined as diamine-CAN (Fig. R16b). This diamine-CAN exhibited a markedly lower elongation at break (331%) than its linear precursor (621%) (Fig. R16c). Crucially, when TBAI was added into this network, the elongation at break

was dramatically restored to 629% (Fig. R16c). This sequence of changes—reduced ductility upon crosslinking and its recovery upon incorporating an ionic plasticizer—directly demonstrates that ionic plasticization, rather than dynamic bond breakage, is the dominant mechanism for the high elongation.

To gain deeper mechanistic insights into this cationic plasticization, we conducted molecular dynamics (MD) simulations. A comparative analysis between the CO₂-derived CAN (with TBA⁺) and the control system (with H⁺ replacing TBA⁺) revealed several fundamental distinctions: (i) a substantially weakened ionic association, as evidenced by a significantly right-shifted first peak in the radial distribution function (RDF) between TBA⁺ and the polymeric anions (Fig. R17a); (ii) markedly enhanced cation mobility, reflected in a diffusion coefficient (D) 2.1 times greater than that of the control system (Fig. R17b-c); (iii) a considerably expanded polymer free volume fraction (FFV, Fig. R17d-e), providing more space for segmental motion; and (iv) dramatically accelerated relaxation dynamics of the polymer chains, as demonstrated by a significantly increased mean-squared displacement (MSD) over time (Fig. R17f-g). These findings collectively demonstrate that the bulky, flexible TBA⁺ cations effectively screen electrostatic interactions, enhance ion mobility, expand free volume, and lubricate the polymer chains, thereby establishing cationic plasticization as the molecular origin of the enhanced ductility and toughness.

Following your recommendation, we also performed cyclic tensile tests on the CO₂-derived CAN, the diamine-crosslinked network, and the plasticized diamine network (Fig. R18). The results revealed pronounced hysteresis and residual strain in all three systems. While this confirms significant plastic deformability, these data alone cannot uniquely identify the underlying molecular mechanism.

In summary, our combined experimental and simulation evidence robustly demonstrates that the exceptional elongation is principally governed by counter-cation plasticization, not by the breaking of CO₃²⁻ crosslinks during stretching. All MD simulation data and corresponding discussion have been added to the revised manuscript (Manuscript: Page 4; Supporting Information: Pages 18-20).

Fig. R16 | **a**, Tensile stress-strain curves of P(VTFP-MA)-20 and P(VTFP-MA)-20 with TBAI additive. **b**, Diamine-CAN were synthesized from P(VTFP-MA)-20 and 1,3-diaminopropane. **c**, Tensile stress-strain curves of P(VTFP-MA)-20, diamine-CAN and diamine-CAN with TBAI additive.

Fig. R17 (newly added Fig. 3i-l and Supplementary Figs. 32-33) | Molecular dynamics (MD) simulations of the counter-cation plasticization mechanism. **a**, Radial distribution function (RDF) between cations and polymeric anions, indicating ionic association strength. **b-c**, Mean-squared displacement (MSD) of counter-cations (**b**) and their corresponding diffusion coefficients (**c**). **d-e**, Polymer free volume fraction (FFV) of the CANs. **f-g**, MSD of polymer backbone (**f**) and their corresponding diffusion coefficients (**g**).

Fig. R18 | Cyclic tensile test curves of diamine-CAN (**a**), diamine-CAN with TBAI additive (**b**) and CO₂-derived CAN (**c**).

Comment 5: P. 3, Fig. 3g: The authors indicate that the mechanical properties of the CO₂-derived CANs decrease with increasing alkyl chain length of the alkylammonium hydroxide. Please describe the cause of this result.

Response: Thank you for this valuable question. Building upon the cationic plasticization mechanism elucidated in our response to Comment 4, we have now provided a detailed explanation for this structure-property trend in the revised manuscript, supported by molecular dynamics (MD) simulations.

Our MD simulations reveal that the systematic mechanical variation originates from a progressive

attenuation of the cationic plasticizing effect as the alkyl chain shortens from TBA⁺ to TMA⁺ and further to the rigid BnMA⁺. This conclusion is supported by several key observations (**Fig. R17 in Comment 4**): radial distribution function (RDF) analysis shows a progressive left-shift of the first peak for the cation–polymer interaction along the series TBA⁺ → TPA⁺ → TEA⁺ → TMA⁺ → BnMA⁺ (Fig. R17a), indicating a decreasing ion-pair distance and thus stronger electrostatic interactions. This strengthened ionic association leads to a concomitant decrease in cation mobility (Fig. R17b-c) and a reduction in polymer free volume (Fig. R17d-e). As a result, the segmental motion of the polymer chains becomes more restricted (Fig. R17f-g), thereby shifting the macroscopic mechanical behavior from ductile to rigid and strong.

All relevant MD simulation data and the corresponding discussion have been incorporated into the revised manuscript (Manuscript: Page 4; Supporting Information: Pages 18-20).

Comment 6: P. 3, Fig. 3h: The authors proposed that the mechanical properties of CO₂-derived CANs are widely tunable, ranging from flexible elastomers to rigid plastics. However, this tunability is mainly due to the characteristics of the polymers used for copolymerization. Therefore, I believe that the advantage of the CO₂-derived CAN may be diminished when rigid EMA and MMA are used. Please compare the mechanical properties before and after CO₂ crosslinking for copolymers with BuMA, EMA, and MMA. If any advantages of CO₂ crosslinking are observed, they should be clearly described.

Response: Thank you for this insightful comment. We fully agree that the intrinsic rigidity of comonomers like EMA and MMA narrows the relative margin for mechanical enhancement via CO₂-derived CO₃²⁻ crosslinking, compared to flexible comonomers. Our comparative analysis demonstrates that CO₃²⁻ crosslinking imparts distinct mechanical improvements across all comonomer systems, even in those based on rigid monomers. For flexible comonomers like BuMA, CO₃²⁻ crosslinking synergistically enhances strength, stiffness, ductility, and toughness, overcoming classical trade-offs inherent in polymer network design, similar to the effect achieved in MA-based CO₂-derived CAN (Fig. R19a-b). For rigid comonomers (EMA, MMA), CO₃²⁻ crosslinking delivers a substantial boost in stiffness, with Young's modulus increasing by over 100% (Fig. R19c-d).

All supplementary data have been incorporated into the revised manuscript (Page 23 in the Supporting Information). We appreciate your valuable feedback.

Supplementary Fig. R19 (newly added Supplementary Fig. S37) Mechanical properties of linear precursor polymers and their corresponding CANs. **a**, Tensile stress-strain curves of P(VTFP-BuMA)-15 and its corresponding CANs. **b**, Comparison of ultimate tensile stress, Young's modulus, toughness, and elongation at break for the samples in (a). **c**, Tensile stress-strain curves of P(VTFP-MMA)-15 and P(VTFP-EMA)-15, alongside their corresponding CANs. **d**, Comparison of Young's modulus and toughness for the samples in (c).

Comment 7: P. 4, Fig. 4: Regarding the experiment shown in Fig. 4, there is no detailed description of the CO₂-derived CAN. Please include an explanation of the prepolymer. Is the prepolymer P(VTFP-MA)-20?

Response: We sincerely thank you for this insightful comment and for pointing out the need for a clearer description. Yes, the prepolymer used to prepare the CO₂-derived CAN is indeed P(VTFP-MA)-20.

To avoid any ambiguity and to make the description more precise, we have now provided a strict definition for "CO₂-derived CAN" in the "Synthesis and Characterization of CANs" section. The revised text explicitly states: "*CO₂-derived CAN (unless otherwise specified, 'CO₂-derived CAN' is defined in this work as the network derived from P(VTFP-MA)-20, TBAH, and ambient air)...*" (Page 4, Line 3). Furthermore, in the end of "Synthesis and Characterization of CANs" section, we have added a clarification to reiterate the specific system: "*Taking CO₂-derived CAN (selected as the representative system for subsequent investigations unless otherwise specified) as an example, ...*" (Page 4, Paragraph 3, Line 8 from the bottom).

We believe these revisions have clearly addressed your concern. Thank you again for helping us improve the clarity of our manuscript.

Comment 8: P. 4, Fig. c and f: The authors conducted self-healing tests at 80°C; however, they also demonstrated that melt processing is possible at a slightly higher temperature of 90°C. In such a case, this reviewer believes that the observed behavior may be attributed to thermoplasticity rather than true self-healing. How would the self-healing performance be affected if the tests were conducted at a lower temperature where melt processing is not possible, such as room temperature?

Response: We thank the reviewer for raising this important point. To directly address this concern, we performed additional self-healing experiments at lower temperatures. The results demonstrate that efficient healing still occurs, with healing efficiencies of 98.2% at 40 °C and 53.2% at 25 °C (Fig. R20). These outcomes confirm that the healing capability is intrinsic and not attributable to thermoplastic flow. All supplementary data and expanded discussion are now incorporated into the revised manuscript (Manuscript: Page 5; Supporting Information: Page 28).

Fig. R20 (newly added Supplementary Fig. S47) | Low-temperature self-healing performance. a–b, Stress-strain curves of the repaired samples after different healing durations under 2 MPa at 40 °C (a), and the corresponding ultimate tensile stress and healing efficiencies (b). **c–d,** Stress-strain curves of the repaired samples after different healing durations under 2 MPa at room temperature (c), and the corresponding ultimate tensile stress and healing efficiencies (d).

Reviewer #3:

Comments for manuscript "Catalyst-Free Upcycling of Air-Captured CO₂ to Self-Healing Recyclable Polymers under Ambient Conditions" Nature Communications.

The manuscript submitted by Prof. Liang Chen and co-workers deals with the synthesis of covalent adaptable networks using ambient CO₂ as a crosslinker which enable the formation of exchangeable bonds. The manuscript is well-written, the molecular study and the effect of the counter ion on the hemiacetal-like moiety formation is highly interesting. The network characterization and material properties are well performed showing excellent mechanical properties and good (re)processability. The manuscript deserves publication after considering the following points especially further studies on rheological characterizations should be performed prior publication.

Response: We sincerely appreciate your positive assessment and insightful suggestions, especially regarding the swelling tests and rheology. All raised points have been incorporated into the revised manuscript, with our detailed responses provided below.

Comment 1: Network design. Can the authors discuss the choice of (co)polymer backbone structure? Why they choose to copolymerize MMA with functionalized styrene and not a fully styrenic polymer backbone?

Response: Thank you for this insightful question regarding the backbone design. Our choice to copolymerize MMA with functional monomer 4-vinyltrifluoroacetophenone (VTFP), rather than using a fully styrenic backbone, was primarily driven by mechanical considerations.

Our initial attempt using a styrenic system—poly(4-vinyltrifluoroacetophenone-co-styrene) [P(VTFP-PS)-18, VTFP/PS = 1/18 mol/mol, $M_n = 56.9$ kDa, $D = 1.89$]—resulted in a material that was exceedingly brittle. Both the linear copolymer precursor and the corresponding CAN fractured during hot-pressing and could not form coherent films (**Fig. R21**). The incorporation of MMA was crucial to resolve this brittleness. The MMA segments impart enhanced ductility and film-forming ability to the dynamic network, enabling the preparation of robust, free-standing films suitable for further characterization and application.

Fig. R21 | Photographic evidence of the mechanical brittleness in P(VTFP-PS)-18 (**a**) and its corresponding CAN (**b**), as demonstrated by their fragmentation upon demolding after hot-pressing.

Comment 2: Swelling tests in good solvents (e.g. THF or CHCl₃) should be performed for each materials. My issue is that the crosslinked nature has not be evidenced either by gel content measurement or DMA.

Response: Thank you for raising this important point. In response, we have carried out comprehensive swelling tests and DMA characterization, which provide clear evidence for the crosslinked nature of our materials, as summarized below:

Swelling Tests of the CO₂-derived CAN: Swelling experiments conducted in various solvents—including tetrahydrofuran (THF), chloroform (CHCl₃), toluene, n-hexane, and ethyl acetate (EA)—demonstrate the material's insolubility (Fig. R22), confirming its crosslinked nature. Quantitative

analysis revealed high gel fractions of 93.3%, 91.2%, 96.5%, 98.8%, and 92.7% in the respective solvents.

Generalization to All CANs: These swelling studies were extended to all synthesized CANs, consistently revealing excellent solvent resistance, with gel fractions in THF exceeding 90% in every case (Fig. R23). These results unambiguously confirm the universal presence of a crosslinked network across the series.

Dynamic Mechanical Analysis (DMA): Further validation comes from temperature-sweep DMA. As illustrated in Fig. R24, the CO₂-derived CAN exhibits a distinct rubbery plateau in the storage modulus (E') above its T_g , in striking contrast to the linear polymer, where E' drops continuously with increasing temperature. This sustained modulus is a key indicator of a dynamic covalent network. Moreover, the crosslink density (ν) was calculated as 79.1 mol·m⁻³ using the equation $\nu = E'_r/[3R(T_g + 50)]$.

Together, these findings conclusively verify the crosslinked structure of all CANs. All relevant data and corresponding discussions have been incorporated into the revised manuscript (Manuscript: Pages 3-4; Supporting Information: Pages 22-23). We appreciate your valuable insight.

Fig. R22 (newly added Fig. 3e) | Gel fraction and swelling ratio of the CO₂-derived CAN in various solvents.

Fig. R23 (newly added Supplementary Figs. S34a, S35a and S36a) | **a**, Gel fraction and swelling ratio of the CANs with different counter-cations. **b**, Gel fraction and swelling ratio of the CANs with different MA-to-VTFP molar ratio. **c**, Gel fraction and swelling ratio of the CANs with different comonomers.

Fig. R24 (newly added Fig. 3g) | DMA curves of P(VTFP-MA)-20 and CO₂-derived CAN.

Comment 3: Rheology experiments to be performed. Stress relaxation experiments presented by the authored showed fast relaxation (between 10 to 100s). Small amplitude oscillatory shear experiments are generally preferred to characterize rapid relaxation (see e.g. *ACS Polym. Au* **2025**, *5*, 3, 214-240). It will also confirm the nature of the exchange mechanism (associative or dissociative) by monitoring the evolution of the storage modulus with increasing temperature. In addition, we generally performed stress relaxation experiments at $T > T_g + 50^\circ\text{C}$ to separate the segmental relaxation induce by the glassy to rubbery transition and the relaxation induce by the exchange reaction. Non-normalized data should be also added in the supplementary material.

Response: We sincerely appreciate your insightful suggestions. As suggested, we have performed the following additional experiments:

(1) Frequency-sweep SAOS experiments: Following your guidance, we conducted frequency-sweep small amplitude oscillatory shear (SAOS) at various temperatures (Fig. R25a). The relaxation time (τ) was determined from the crossover frequency (ω_c) where $G' = G''$ ($\tau = 1/\omega_c$). An Arrhenius plot of $\ln(\tau)$ versus $1000/T$ showed a linear correlation (Fig. R25b), confirming the vitrimeric nature of the networks. The activation energy (E_a) for the bond exchange was calculated to be 111.9 kJ/mol. Subsequently, the master curve was constructed by applying the time-temperature superposition (TTS) principle to the SAOS data at a reference temperature of 60 °C (Fig. R25c). The horizontal shift factors (a_T) from the TTS procedure were analyzed via the Arrhenius equation, yielding an activation energy of 109.3 kJ/mol (Fig. R25d). This value is consistent with that from the crossover frequency method, validating the reliability of our rheological analysis.

Fig. R25 (newly added Fig. 4a-d and Supplementary Fig. 44) | a-b, Frequency-sweep SAOS measurements at various temperatures **(a)** and corresponding Arrhenius plot **(b)**. **c-d,** Master curve of CO₂-derived CAN based on TTS of SAOS experiments at a reference temperature of 60 °C **(c)**, and the corresponding Arrhenius plot of horizontal shift factors (a_T) versus $1000/T$ **(d)**. **e,** Temperature-dependent rheological test of the CO₂-derived CAN. **f,** *In situ* variable-temperature FTIR spectra of CO₂-derived CAN.

(2) Temperature-sweep rheological test: To confirm the nature of the exchange mechanism, we first performed temperature-sweep rheological experiments. The results show a sharp, nearly three-

order-of-magnitude decrease in G' accompanied by a distinct G'/G'' crossover at approximately 85 °C (Fig. R25e). This behavior is a hallmark of a dissociative exchange mechanism, as manifested by the sudden loss of network integrity upon heating.

(3) *In situ variable-temperature FTIR*: To provide molecular-level validation, we conducted *in situ* variable-temperature FTIR spectroscopy (Fig. R25f). The spectra directly demonstrate the thermally induced cleavage of the CO_3^{2-} -bridged dynamic covalent bonds upon heating and their subsequent reformation upon cooling, offering unequivocal evidence for the dissociative exchange process.

All new data and corresponding discussion have been incorporated into the revised manuscript (Manuscript: Pages 4-5; Supporting Information: Pages 26-27). We are grateful for these valuable comments, which have significantly strengthened our mechanistic analysis.

Comment 4: In addition, creep experiments especially at room temperature should be performed to determine if the material mechanical properties are stable under ambient conditions. DMA analysis to further confirm the CAN features (presence of a rubbery after T_g).

Response: Thank you for this valuable suggestion. As recommended, we conducted tensile-mode DMA creep tests at room temperature. The results reveal significant viscoelastic flow, with the residual strain reaching 49% (Fig. R26). This pronounced creep deformation indicates the activation of dynamic covalent bond exchange even under ambient conditions. While this suggests that long-term dimensional stability under sustained load may require further optimization, it also underscores the material's potential for low-temperature self-healing applications. To validate this, we performed quantitative self-healing assessments at low temperatures. The network achieved excellent healing efficiencies of 53.2% at 25 °C and 98.2% at 40 °C within 24 hours (Fig. R27), consistent with the room-temperature bond exchange dynamics observed in the creep tests.

In addition, as also addressed in our response to **Comment 2**, temperature-sweep DMA experiments confirm a well-defined rubbery plateau above T_g , verifying the crosslinked network structure.

All relevant data, including low-temperature self-healing performance, DMA curves, and associated discussions, have been incorporated into the revised manuscript (Manuscript: Page 3, Page 5; Supporting Information: Page 28).

Fig. R26 | Creep curves of CO_2 -derived CAN at room temperature.

Fig. R27 (newly added Supplementary Fig. S47) | Low-temperature self-healing performance. a–b, Stress-strain curves of the repaired samples after different healing durations under 2 MPa at 40 °C (a), and the corresponding ultimate tensile stress and healing efficiencies (b). **c–d,** Stress-strain curves of the repaired samples after different healing durations under 2 MPa at room temperature (c), and the corresponding ultimate tensile stress and healing efficiencies (d).

Point-by-Point Response to Reviewers' Comments

Reviewer #1:

The manuscript entitled "Upcycling of Atmospheric CO₂ to Self-Healing Recyclable Polymers under Ambient Conditions" describes a strategy for producing cross-linked polymers with self-healing and closed-loop recyclability directly from atmospheric CO₂ under catalyst-free conditions, presenting a valuable pathway toward green and reversible crosslinking in polymer science. This study leverages the rapid absorption of atmospheric CO₂ by organic bases such as TBAC, converting it into CO₃²⁻ which enable the dynamic reversible cross-linking of P(VTFP-MA)-20. The resulting covalent network exhibits excellent self-healing properties, recyclability, and favorable mechanical performance.

Overall, the referee believes that the manuscript deserves publication in Nature Communications after the identified issues are addressed.

Response: Thank you for your supportive feedback and insightful suggestions. All raised points are addressed as follows.

Here are the minor comments the reviewer suggests:

1. Regarding Fig. 3c, it would be more informative to plot the pH profile of the TBAH solution over time. The current format, which only labels the initial and final pH values, could be misleading to readers.

Response: Fig. 3c has been revised into a dual-y-axis plot showing both exhaust CO₂ and TBAH pH over time, with an updated caption (please see below). We appreciate this helpful suggestion.

Fig. R1 (replaced with Fig. 3c) | CO₂ concentration in the exhaust stream and pH of the solution over time during the bubbling of ambient air.

2. While the model reactions effectively prove CO₃²⁻ reacts with TFP, the study does not address whether HCO₃⁻ could also participate in or interfere with this reaction. During the cross-linking of the linear polymer, ambient air was bubbled through the TBAC solution. This process may inevitably produce HCO₃⁻. It is important to clarify whether the presence of HCO₃⁻ competes with CO₃²⁻ and adversely affects the efficiency of the cross-linking reaction.

Response: Thank you for this insightful comment. As established in our previous data, the TBAC prepared by bubbling ambient air contains ~94.1% CO₃²⁻ alongside ~5.9% HCO₃⁻. To determine whether this trace HCO₃⁻ impurity interferes with the subsequent cross-linking or final network properties, we conducted the following two experiments.

(1) **Model Reaction Probes Chemoselectivity.** To probe inherent reactivity, a stoichiometric mixture of the model compound α,α,α -trifluoroacetophenone (TFP) and tetrabutylammonium hydrogen carbonate (TBAHC) was stirred in CD₃CN at room temperature for 24 hours. ¹H, ¹⁹F, and

^{13}C NMR analysis of the reaction mixture showed no detectable reaction, with spectra identical to those of the starting materials (**Fig. R2**). This stands in marked contrast to the quantitative reaction observed between TFP and CO_3^{2-} under identical conditions. This confirms the high selectivity for CO_3^{2-} over HCO_3^- , which is attributable to the superior nucleophilicity of CO_3^{2-} .

(2) Effect of Trace HCO_3^- on Cross-linked Polymer Properties. To further evaluate the practical impact, a control network was synthesized using pure TBAC (no prior air CO_2 exposure) and comprehensively compared with the CO_2 -derived CAN. The two networks exhibited negligible differences in both glass transition temperature (T_g) and mechanical performance—including Young's modulus, ultimate tensile stress, toughness, and elongation at break (**Fig. R3**).

Together, these results demonstrate that the trace HCO_3^- impurity neither participates in the cross-linking reaction nor impairs the final thermomechanical properties of the networks. All corresponding data and discussion have been incorporated into the revised manuscript (**Main text: Page 3; SI: Pages 17-19**).

Fig. R2 (newly added Supplementary Fig. 27) | Comparison of NMR spectra (in CD_3CN) for TFP, TBAHC, and the reaction mixture. **a**, ^1H NMR, **b**, ^{13}C NMR, and **c**, ^{19}F NMR.

Fig. R3 (newly added Supplementary Fig. 28) | Comparison of thermal and mechanical properties between the CO_2 -derived CAN and the network cross-linked with pure TBAC. **a**, DSC curves. **b**, Representative tensile stress-strain curves. **c**, Comparison of Young's modulus, ultimate tensile stress, toughness, and elongation at break.

3. Referee believes that the statement "the urgent need for novel catalyst-free dynamic chemistries" in the introduction part is inaccurate, since many dynamic covalent bonds can undergo efficient exchange under thermal activation without catalyst. The reviewer recommends that the authors revise this statement to ensure accurate expression.

Response: We have revised the sentence on **Page 1**, removing the claim regarding "catalyst-free" chemistry. It now reads: "...*These limitations highlight the need for novel dynamic chemistries that simultaneously achieve superior reprocessability, chemical recyclability, and balanced mechanical properties.*"

4. The introduction on the up-to-date progress in CO₂ utilization could be strengthened. It would be beneficial to cite recent achievements in this field (see e.g. *Nat. Commun.*, 15, 6605 (2024)) to provide a more comprehensive background of the present work.

Response: We've incorporated the suggested reference (**Ref. 32**) into the Introduction, enhancing the background on recent CO₂ utilization progress relevant to our study. Thank you for the helpful suggestion.

5. Additionally, there are a few problems that should be dealt with before published. For example: (1) The letter labels (j, k, l) in the caption of Fig. 3 should be bolded and must conform to standard typesetting conventions.

Response: The letter labels (j, k, l) in the Fig. 3 caption have been bolded and standardized. Thank you for pointing this out.

(2) In Fig. 5b, a space should be inserted between the axis label name and the following parentheses.

Response: Noted and corrected. We have also verified and corrected this issue in all other figures.

Reviewer #2:

The authors have conducted additional experiments and have appropriately addressed the reviewer's questions. I recommend accepting this paper in its current version.

Response: We appreciate the reviewer's support and recommendation for publication.

Reviewer #3:

The revised version of the manuscript entitled "Upcycling of Atmospheric CO₂ to Self-Healing Recyclable Polymers under Ambient Conditions" includes all the suggested modifications especially about the rheological behavior of these original materials. I therefore recommend publication of this manuscript.

Response: We appreciate the reviewer's recommendation for publication.